# A genetic toolkit for the human gut bacterium *Mediterraneibacter gnavus* identifies capsular polysaccharides as a competitive colonization factor

Nozomu Obana ®[1,2] ✉, Gaku Nakato[3,4], Nobuhiko Nomura[2,5,6] & Shinji Fukuda ®[1,2,3,4,7] ✉

*Mediterraneibacter gnavus* is a human symbiotic gut bacterium whose abundance often increases in patients with various diseases, such as active inflammatory bowel disease (IBD). However, the genetic factors governing its gut colonization and pathogenicity remain elusive due to the lack of genetic modification systems. In this study, we developed several genetic tools for *M. gnavu*s, including a shuttle vector, an inducible promoter, fluorescent reporters, and systems for gene disruption and deletion. Using these genetic tools, we constructed mutants for six of the eight sortase-encoding genes in *M. gnavus* ATCC 29149 and identified those involved in the surface presentation of capsular polysaccharide (CPS) and superantigen-like proteins. We also identified a CPS biosynthetic gene cluster adjacent to the sortase gene and demonstrated that CPS production is crucial for competitive colonization in germ-free mouse intestines. Notably, CPS production was inversely correlated with inflammatory activity, and CPS cluster-positive strains were more prevalent in healthy individuals than in Crohn's disease patients. These findings suggest that CPS contributes to the modulation of inflammation and pathogenesis. This study highlights the potential of precise gene-modification systems to uncover genetic determinants of intestinal colonization and pathogenesis in gut bacteria.

The human gut microbiota plays a pivotal role in maintaining host health and modulating disease processes[1]. Dysbiosis—an imbalance in microbiota composition—has been implicated in various conditions, including inflammatory, metabolic, and neurodevelopmental disorders[2,3]. While metagenomic and metabolomic approaches have elucidated disease-associated microbial patterns, the causal roles of specific bacterial genes in host pathophysiology remain poorly understood[4]. This knowledge gap stems in part from the limited availability of genetic tools for non-model gut bacteria, which constitute the majority of the gut ecosystem. Recently, the functional analysis of non-model gut bacteria, along with the development of a gene modification system for specific bacteria, has been expanding and attracting attention[5].

[1]Transborder Medical Research Center, Institute of Medicine, University of Tsukuba, Tsukuba, Japan. [2]Microbiology Research Center for Sustainability (MiCS), University of Tsukuba, Tsukuba, Japan. [3]Gut Environmental Design Group, Kanagawa Institute of Industrial Science and Technology, Kawasaki, Japan. [4]Innovative Microbiome Therapy Research Center, Juntendo University Graduate School of Medicine, Tokyo, Japan. [5]Institute of Life and Environmental Sciences, University of Tsukuba, Tsukuba, Japan. [6]Tsukuba Institute for Advanced Research, University of Tsukuba, Tsukuba, Japan. [7]Institute for Advanced Biosciences, Keio University, Tsuruoka, Japan. ✉e-mail: obana.nozomu.gb@u.tsukuba.ac.jp; sfukuda@sfc.keio.ac.jp

*Mediterraneibacter gnavus* (formerly *Ruminococcus gnavus*) is a symbiont in the human gut which has been linked to various diseases, including inflammatory bowel disease (IBD), irritable bowel syndrome (IBS), and allergy[6–9]. *M. gnavus* is frequently detected in the intestines of patients with these diseases[10] and its increased abundance is often positively correlated with the severity of symptoms. This implies that *M. gnavus* may induce inflammation by blooming in the intestines or alternatively may preferentially thrive in the inflammatory environment. Since *M. gnavus* has been reported to produce inflammatory polysaccharides[7] and degrade mucus layers[11], this bacterium is likely to induce inflammatory responses in the host. Several studies have reported strain-specific pro-inflammatory properties[12,13], suggesting that certain *M. gnavus* genes in specific strains may contribute to mucosal colonization and inflammation, and that not all strains may be equally pathogenic. However, the genetic determinants underlying such phenotypic diversity remain largely unknown.

Given the association between *M. gnavus* abundance and inflammation, understanding the molecular and genetic mechanisms of intestinal colonization in *M. gnavus* is crucial. Bacteriocin production[14], mucin degradation activity[15], adhesin binding to mucins[16] and biofilm formation[17] have been previously identified as strategies for intestinal colonization in *M. gnavus*. However, the number of experimentally validated colonization factors is still limited. Moreover, the strain-specific nature of certain phenotypes, such as capsular polysaccharide (CPS) production and mucin degradation, underscores the need for comparative genomic analysis, as well as functional validation using isogenic mutant strains[18,19]. Such approaches, however, require a robust genetic manipulation system, something that has been notably lacking in *M. gnavus*.

In this study, we develop and apply a suite of genetic tools for *M. gnavus*, including a shuttle vector, inducible promoters, fluorescent markers, and gene disruption systems. Using these tools, we uncover the roles of specific sortase enzymes in cell-surface protein localization and identify a previously uncharacterized gene cluster responsible for CPS biosynthesis. Functional studies using germ-free mice demonstrate that CPS production is essential for competitive colonization and is inversely correlated with inflammatory potential. Furthermore, our genomic analyses reveal that CPS-producing strains are more prevalent in healthy individuals than in patients with Crohn's disease, suggesting that CPS plays a role in both colonization and disease modulation. In this study, we develop a genetic toolkit that enables the functional dissection of specific sortase genes and capsular polysaccharide biosynthesis pathways in *M. gnavus*, a human gut symbiont. This work provides insights into how strain-specific genetic variation shapes surface structure and may influence host interactions and disease relevance.

## Results

### Development of genetic tools in *M. ganvus*

Precise gene-modification systems are necessary for understanding gene functions and the causal relationships between bacteria and diseases. To facilitate genetic modifications for *M. gnavus*, we began by creating an *E. coli-M. gnavus* shuttle vector containing an origin suitable for *M. gnavus* (Fig. 1a and Supplementary Fig. 1; see Methods). Furthermore, to find the inducible promoter for *M. gnavus*, we cloned cumate-[20], lactose-[21], tetracycline-[22], and xylose-inducible promoters[23] upstream of mScarlet-I, red fluorescent protein gene[24] (Supplementary Fig. 2). The cumate-inducible promoter showed constitutive expression in pNORg11, in which the *cymR* gene encoding the repressor is under the control of a modest promoter, $P_{xyl}$. When the *cymR* gene is under the control of the strong constitutive promoter $P_{fdx}$, mScarlet-I expression in pNORg12 is inducible by a high concentration of cumate in *M. gnavus* (Fig. 1B and Supplementary Fig. 2). However, we observed slight, leaky mScarlet-I expression in the absence of the inducer cumate. To reduce the leaky expression, we introduced a theophylline

riboswitch into the Shine-Dalgarno (SD) sequence in pNORg14, resulting in no apparent leaky expression (Fig. 1a, b). mScarlet-I expression from pNORg14 can be induced in the presence of both cumate and theophylline. A series of plasmids with different expression patterns could facilitate experiments for various target genes, including those that are toxic or difficult to clone into the plasmid.

Using the constitutive expression system in pNORg11, we generated strains expressing fluorescent proteins, which differ in excitation and emission spectra, photostability, and maturation time (Fig. 1c). Because *M. gnavus* are usually cultured anaerobically, but these fluorescent proteins require oxygen for fluorophore maturation, we exposed the cells to oxygen for 2 h. Exposure to oxygen did not lead to obvious cell lysis of *M. gnavus*. After exposure to oxygen, we can observe these cells by fluorescent imaging, indicating that these fluorescent proteins allow for the fluorescent labeling of *M. gnavus* cells and the construction of promoter-reporter strains.

Next, we attempted to improve the gene disruption and deletion systems in *M. gnavus*. The Clostron system can be applied to constructing gene disruptants in *M. gnavus*[15,25]. We previously built a plasmid of the Clostron system for *C. difficile*[26] and confirmed its availability for *M. gnavus* (Fig. 1d). However, this system utilizes the erythromycin resistance gene to select the mutant strain, which limits the construction of multiple gene mutant strains. To achieve more precise gene modification, we developed a plasmid carrying CRISPR/Cas9 (Fig. 1e). For proof of concept, we attempted to construct the double mutant strain of superantigens, which can bind to human IgG (hIgG) and IgA (hIgA), encoded by the genes *ibpA* and *ibpB*[27]. Using the Clostron and CRISPR/Cas9 systems, we successfully constructed the *ibpA* and *ibpB* double mutant strain (Supplementary Fig. 3a–e) as evidenced by the absence of IgA-bound protein detected by Western blotting using hIgA (Fig. 1f). We performed a whole genome sequencing analysis of this strain, showing that there are no off-target effects from the CRISPR/Cas9 system (Supplementary Fig. 3f).

Taken together, we developed plasmid-based genetic tools to construct various gene-modified strains of the human gut symbiont *M. ganvus* and likely other closely related bacteria, such as other members of the Lachnospiraceae family.

### *M. gnavus* possesses multiple sortases

To identify the genetic factors that contribute to intestinal colonization in *M. gnavus* using the genetic tools, we selected sortase genes, which encode the enzyme responsible for mediating protein anchoring to cell walls, a process essential for proper cell wall protein localization. Sortase has divergent roles associated with recognition by the host immune system and colonization in several bacteria[28,29]. *M. gnavus* ATCC 29149, a type strain, has eight putative sortase genes, which we named *srtB1-6* and *srtC1-2* based on a phylogenetic analysis of their amino acid sequences (Supplementary Fig. 4a). Out of 8 putative sortase genes, we successfully created mutant strains for six genes (*srtB1-5* and *srtC1*: Supplementary Fig. 5a and b). Genome resequencing analysis revealed that the *srtC2* and *srtB6* genes may be duplicated in our stock of the *M. gnavus* strain, which may explain why we were unable to disrupt these genes (Supplementary Fig. 5c).

### SrtB3 is necessary for the cell wall localization of superantigen proteins

Superantigens, which are cell-surface proteins widely conserved in *M. gnavus*, can bind to human IgA and stimulate B cells, prompting us to test the localization of superantigen proteins in the sortase mutants. The C-terminal domain of Ibp proteins contains the SPXTG motif, a feature also observed in sortase-processed cell wall-anchored proteins[27,30]. To test the effect of sortase genes on superantigen localization, we detected hIgA-bound proteins by western blotting of the cytoplasmic, cell wall, and extracellular fractions of wild-type and mutant strains (Fig. 2a). We confirmed that superantigens were

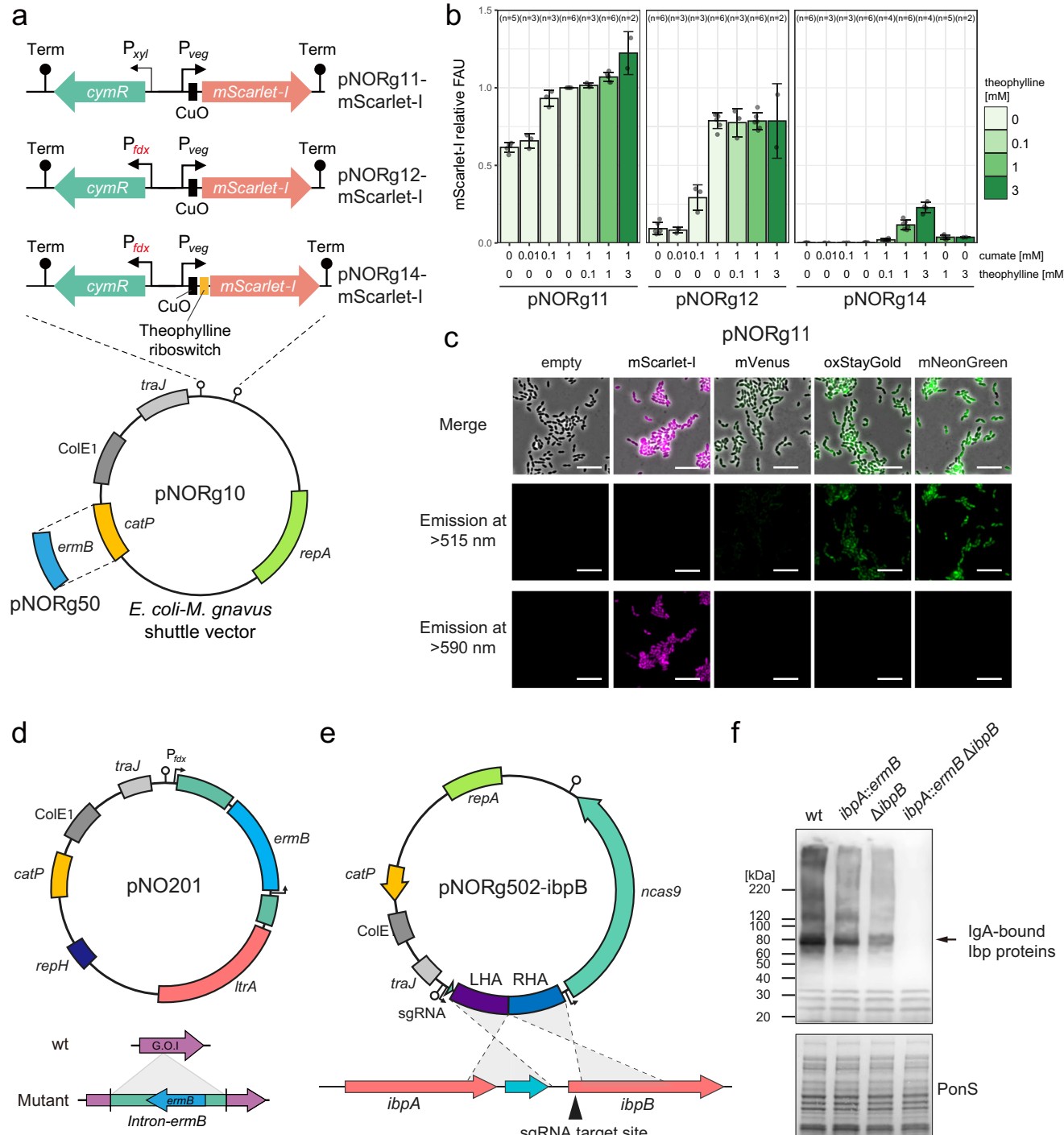

**Fig. 1 | Development of a genetic toolkit for *M. gnavus*. a** Schematics of *E. coli-M. gnavus* shuttle plasmids, which contain constitutive or inducible promoters. **b** Fluorescence reporter analysis. Cells carrying plasmids with mScarlet-I were grown anaerobically with or without inducers, cumate, and theophylline. Fluorescence intensities (FAU) are normalized by the value of cells carrying pNORg11 incubated with 1 mM cumate. Bars represent means ± SD from independent experiments. The number of experimental replicates (*n*) is indicated above each bar. **c** Fluorescence microscope images of the fluorescent strains. *M. gnavus* harboring pNORg11, which carries fluorescent protein genes, was grown at the late-exponential phase and exposed to oxygen for >2 h. Fluorescent proteins are visualized by excitation wavelength filtered with BP 450–490 nm or BP 546/12 and

emission wavelength filtered with LP 515 or LP 590. Merged images with phase contrast microscopy are shown. Bar = 10 μm. **d** A schematic of the plasmid for gene disruptant construction. **e** A schematic of the plasmid for constructing an *ibpB* mutant using CRISPR-Cas9. The plasmid includes an sgRNA targeting the chromosomal *ibpB* locus and a repair template spanning the adjacent upstream and downstream regions of *ibpB*. **f** Western blotting for IbpA and IbpB detection. Cells were grown to the mid-exponential phase. Whole cell lysate prepared from the culture was separated by SDS-PAGE. Each lane contains 0.025 O.D.600 units. IbpA and IbpB proteins were probed with human IgA. For **c**, **f**, similar results were obtained in two independent experiments.

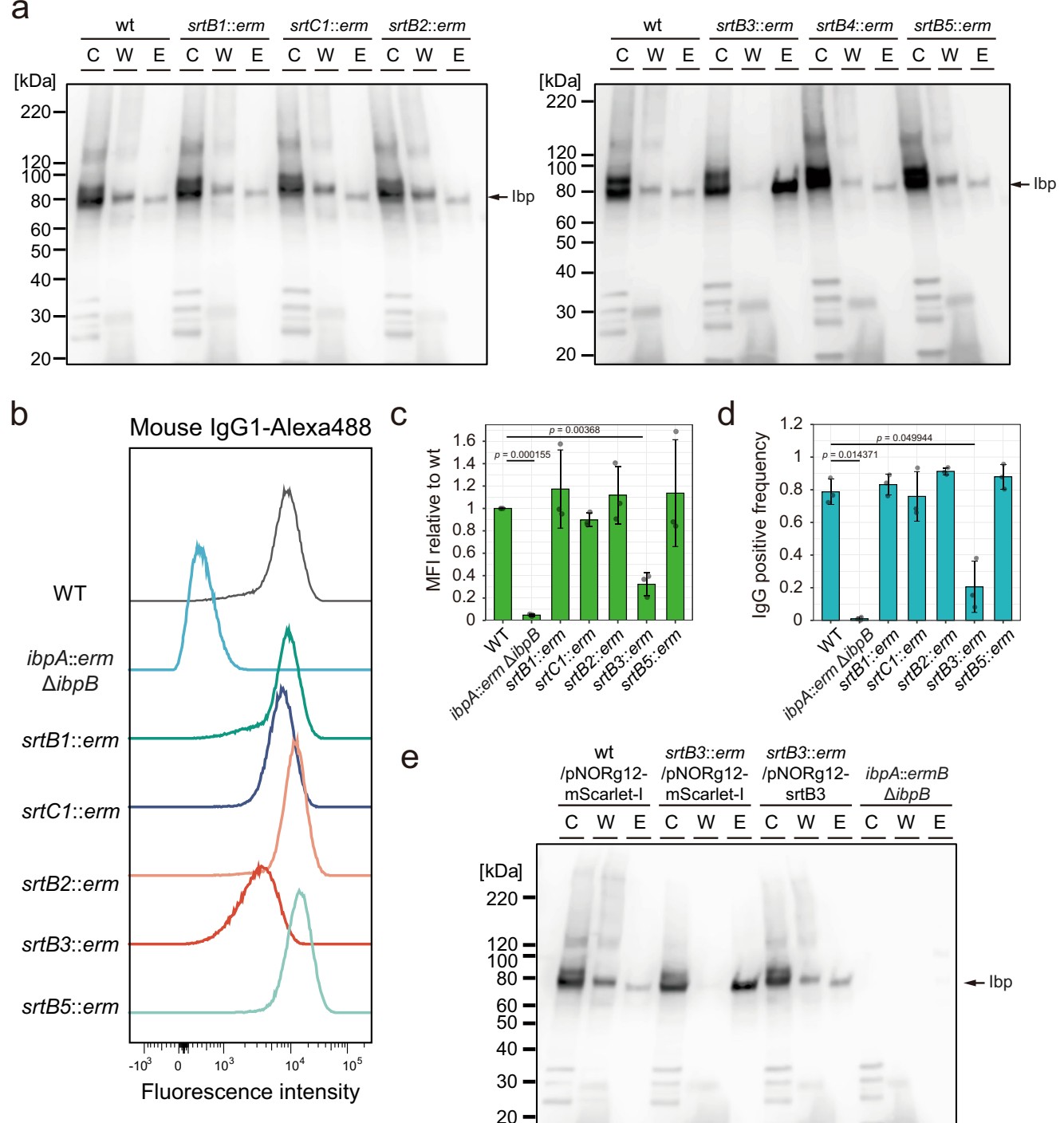

**Fig. 2 | *srtB3* is responsible for cell wall localization of superantigens.**
**a** Subcellular localization of superantigens, IbpA, and IbpB in various sortase gene mutants. Western blotting against hIgA detects superantigens. Proteins (C, cytoplasmic; W, cell wall; E, extracellular) were isolated from the cells grown at the mid-exponential phase. Each lane contains 0.025 O.D.600 units. **b–d** Flow cytometry analysis detecting Mouse IgG-bound *M. gnavus* cells. Cells grown at the late-exponential phase were mixed with Alexa 488-conjugated mouse IgG1. Representative histograms were shown (**b**). Bars represent means ± SD of mean fluorescent intensities (MFI) relative to WT (**c**) and IgG-positive frequencies (**d**),
calculated from 3 independent experiments. Statistical differences were determined using one-sided Welch's t-test with Bonferroni correction for multiple comparisons (*p* < 0.05). Exact *p*-values are shown. **e** Complementation of *srtB3* restores the subcellular localization of the superantigens in *M. gnavus*. Proteins were isolated from cells grown at the mid-exponential phase, which harbor an empty plasmid or the *srtB3*-expressing plasmid. Each lane contains 0.025 O.D.600 units. hIgA was used to detect the superantigens. For **a**, **e**, similar results were obtained in two independent experiments.

detected in cell wall fractions, as predicted, and also detected these proteins in extracellular fractions. In the *srtB3* mutant, superantigens that were localized to the cell wall fractions were abolished and were primarily localized to extracellular fractions (Fig. 2a). To confirm the

loss of superantigen localization to the cell surface in the *srtB3* mutant, we labeled the cells with fluorescence-tagged IgG and analyzed the IgG-bound cells by flow cytometry. The IgG-bound cell decreased in the *srtB3* mutant (Fig. 2b–d). Additionally, complementation by the

*srtB3*-expressing plasmid (pNORg12-srtB3) restored the cell wall localization of Ibp proteins (Fig. 2e). These indicate that the SrtB3 enzyme anchors superantigens to the cell wall in *M. gnavus*.

### *srtB4* is necessary for capsular polysaccharide presentation

Next, since *M. gnavus* is known to produce CPS, which is associated with the induction of inflammatory cytokines[12], we observed CPS production by microscopy using Indian ink staining in the sortase mutants. We confirmed CPS production as a "halo" around the cells when observed after mixing the culture with Indian ink. The *srtB4* mutant did not exhibit a halo around the cells (Fig. 3a). The *srtB4* mutant formed tightly packed cell pellets after centrifugation, whereas the wild-type strain and the other mutants formed a less compact pellet (Fig. 3b, bottom panel). CPS production has been reported to cause cell pellets to become loose in various bacteria[12]. Furthermore, we detected acid-polysaccharides, the typical main constituents of CPS, in each mutant strain by SDS-PAGE using alcian blue staining[31]. The smear signals indicating polysaccharides in the cell wall fractions were significantly reduced in the *srtB4* mutant strain (Fig. 3b top panel). Finally, we observed fibrous structures surrounding the wild-type cells but not the *srtB4* mutant by transmission electron microscopy (Fig. 3c).

To further confirm the responsibility of *srtB4* in CPS presentation, since *srtB4* is located within a gene cluster, we constructed the *srtB4* mutant strain using homologous recombination with the CRISPR-Cas9 system (Fig. 3d). We confirmed the deletion of the *srtB4* gene in the established mutant strain by diagnostic PCR (Fig. 3e). The resulting Δ*srtB4* did not produce a substantial amount of acid-polysaccharides, and the cell pellets formed after centrifugation were tightly packed (Fig. 3f). Furthermore, we confirmed that the complementation by *srtB4*-expressing plasmid restored CPS presentation (Fig. 3f, g). These results clearly show that *srtB4* is necessary for the surface presentation of CPS in *M. gnavus*.

### *srtB4* mutant loses competitive fitness during murine intestinal colonization

CPS production is frequently associated with the colonization ability of bacteria in the host[32]. To assess the involvement of sortase genes in competitive fitness for intestinal colonization, we measured the competitive advantage of wild-type (wt) strains against the sortase mutants during colonization into germ-free mouse intestines. We mixed each sortase mutant with the wild type at a 1:1 ratio and co-inoculated it into germ-free mice. After 1 or 2 weeks of inoculation, we measured the colony-forming units (CFU) of each strain in feces to calculate the competitive index (for detailed procedures, see "Methods") (Fig. 4a). Most mutant strains displayed a slightly decreased competitive index after two weeks of colonization. Among them, the *srtB4* mutant was remarkably outcompeted by the wild type at one week of colonization and was undetectable after two weeks. We also measured the in vitro growth rate of sortase mutant strains. The in vitro growth rate of the *srtB4* mutant is comparable to that of the wild type (Fig. 4b). In addition, the wild type did not outcompete the *srtB4* mutant strain under in vitro co-culture conditions (Fig. 4c). These results indicate that *srtB4*-encoding sortase is indispensable for competitive colonization of *M. gnavus* ATCC 29149 in the murine intestine.

### Identification of a gene cluster responsible for CPS production in *M. gnavus* ATCC 29149

Since sortase is unlikely to catalyze the direct production of CPS, we sought to identify genes involved in CPS biosynthesis. We found that a gene cluster containing the *srtB4* gene is composed of genes predicted to be involved in CPS biosynthesis, including sugar biosynthesis enzymes, glycosyltransferases, and a peptidoglycan attachment enzyme (Fig. 5a). Furthermore, we confirmed promoter activity in several intergenic regions in the gene cluster in our experimental

conditions using the fluorescent reporter system (Supplementary Fig. 6). We chose several genes putatively encoding nucleotide sugar biosynthesis protein (RGna_11905), glycosyltransferase (RGna_11920), the LytR-CpsA-Psr (LCP) family enzyme (RGna_12040), and bacterial tyrosine phosphatase (RGna_12055) to test if the genes were necessary for CPS production (Fig. 5a). Nucleotide sugar biosynthesis is required to generate the substrates for the reaction of glycosyltransferase, which mediates polysaccharide repeat unit production[33]. The LCP family enzyme is known to catalyze cell wall attachment of CPS[34]. The BY-kinase/phosphatase pair is reported to regulate CPS production[35]. All of the mutants exhibited defective CPS presentation phenotypes as evidenced by microscopic observations, tightly packed cell pellets after centrifugation, and reduced levels of acid-polysaccharides detected in SDS-PAGE (Fig. 5b–d). These results suggest that the gene cluster around *srtB4* is responsible for CPS biosynthesis in *M. gnavus* ATCC 29149. A previous study predicted another gene cluster as CPS biosynthesis genes in the *M. gnavus* ATCC 29149 genome[12]. We also tested CPS presentation in the mutant strain of genes located in this locus (RGna_00045 and RGna_00055, which are putatively encoding glycosyltransferase and LCP family protein, respectively). However, in our experimental conditions, these gene disruptions did not affect the phenotypes tested in our study (Fig. 5b–d).

### CPS biosynthesis genes are necessary for competitive fitness in the murine intestine

We reasoned that if SrtB4-mediated colonization fitness is attributed to CPS presentation to cell surfaces, the disruption of the CPS gene should also result in decreased colonization fitness during an in vivo competition assay against the wild-type strain. We co-inoculated the CPS mutant strains (RGna_11905 and RGna_12040) with the wild-type strain into germ-free mice and measured competitive fitness to test the hypothesis. While the RGna_00045 mutant did not exhibit any deficiency in colonization ability, the RGna_11905 and RGna_12040 mutants were rapidly outcompeted by the wild-type strain during intestinal colonization (Fig. 5e). These findings indicate that CPS production, facilitated by its biosynthesis genes, is crucial for competitive colonization in *M. gnavus*. We observed that the in vitro growth rate of RGna_11905 mutant was slightly decreased compared to that of the wild-type strain, but RGna_11905 was not outcompeted by the wild-type strain in the in vitro competition assay (Supplementary Fig. 7 and Fig. 5f). Because RGna_11905 is a putative nucleotide-sugar synthesis gene, it may affect the production of nucleotide-linked sugars necessary for other cell glycans. This is a possible explanation for the mutant strain exhibiting decreased competitive fitness. However, results including RGna_12040, which may only be involved in polysaccharide attachment, are likely to rule out a simple difference in growth rate as the cause of the CPS mutant strains being outcompeted by wild-type in the intestinal environment. Taken together, the CPS produced by the gene products of the identified locus is necessary for competitive fitness during intestinal colonization in *M. gnavus*.

### CPS production in *M. gnavus* is inversely correlated with inflammation activities both in vitro and in vivo

Because CPS is attributed to have various roles in immune recognition or modulation, CPS production in *M. gnavus* may influence the innate immune response of host cells. To determine the influence of CPS production, we incubated murine macrophage-like J774.1 cells with *M. gnavus* wild-type and CPS mutant strains and measured cytokine production from host cells. TNF-α production is stimulated by *M. gnavus* cells and is significantly increased in the presence of the *srtB4* mutant (Fig. 6a). The RGna_11905 mutant also displayed increased TNF-α production compared to wild-type, suggesting that CPS production is inversely correlated with cytokine induction by *M. gnavus* cells (Fig. 6a). To further confirm whether CPS production is involved in intestinal inflammation induced by *M. gnavus*, we prepared *M.*

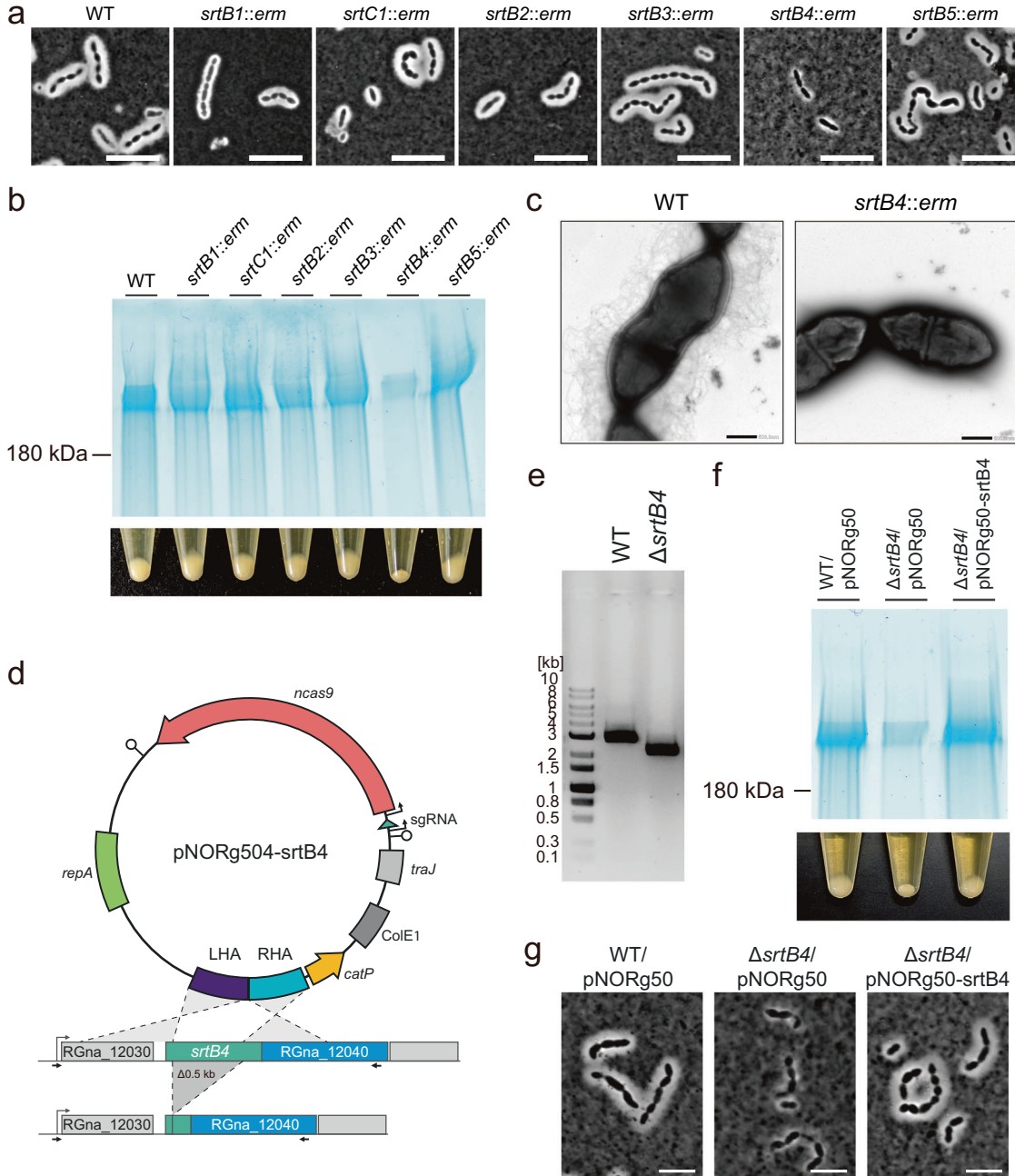

**Fig. 3 | srtB4 is responsible for the surface presentation of capsular poly-saccharide. a** Indian ink staining of each sortase gene mutant. Cells grown over-night were mixed with an equal volume of Indian ink and observed using a phase contrast microscope. Bar = 5 μm. **b** Capsular polysaccharide production detected by SDS-PAGE and loose cell pellets. Capsular polysaccharide fractions collected from cells grown overnight were separated by SDS-PAGE. Capsular polysaccharides were detected by acid polysaccharide staining dye Alcian blue (Upper panel). Each lane contains 0.5 O.D. 600 units. Cell pellets after centrifugation (10,000 × g for 2 min) of cell cultures are also shown in the bottom panel. Cells in which capsular polysaccharides were detected by Indian ink staining and SDS-PAGE represent loosened pellets. **c** Transmission microscopic images. Cells were grown overnight. Bar = 500 nm. **d** Schematics of a plasmid map for constructing the non-polar mutant of the *srtB4* gene. Left and right homology arms (LHA and RHA) are shown

with homologous adjacent regions of *srtB4*. The desired mutant strain lacks approximately 0.5 kb in the internal region of *srtB4*. Allows indicate primers used for diagnostic PCR of the deletion. **e** Confirmation of the deletion by PCR targeting *srtB4* neighboring region. PCR products using the Δ*srtB4* genome as a template DNA are 0.5 kb shorter than those using the wt genome. **f** Capsular polysaccharide production detected by SDS-PAGE and loose cell pellets. Capsular polysaccharide fractions collected from cells harboring the empty or *srtB4*-expressing plasmid. The upper and bottom panels show SDS-PAGE images of capsular polysaccharides and Cell pellets, respectively, which were detected using the same method as in Fig. 2B. **g** Indian ink staining of *srtB4* mutant and *srtB4*-complemented strain. Cells grown overnight were mixed with an equal volume of Indian ink and observed using a phase contrast microscope. Bar = 5 μm. For **a–c, e–g**, similar results were obtained in two independent experiments.

*gnavus* wild-type or *srtB4* mutant-mono-colonized mice and then compared inflammation during administration with dextran-sodium sulfate (DSS) (Fig. 6b). Despite no substantial difference in coloniza-tion levels between wt and *srtB4* mutant in gnotobiotic mice at day 0 and a modestly increased number of *srtB4* mutant strains at 8 days post

DSS treatment (Fig. 6c), we observed more severe body weight loss and increased disease activity index (DAI) in *srtB4* mutant-colonized mice compared to wild-type (Fig. 6d, e). These data suggest that genetic differences in *M. gnavus* colonizing the intestine influence inflammation, and that loss of CPSs may exacerbate its severity.

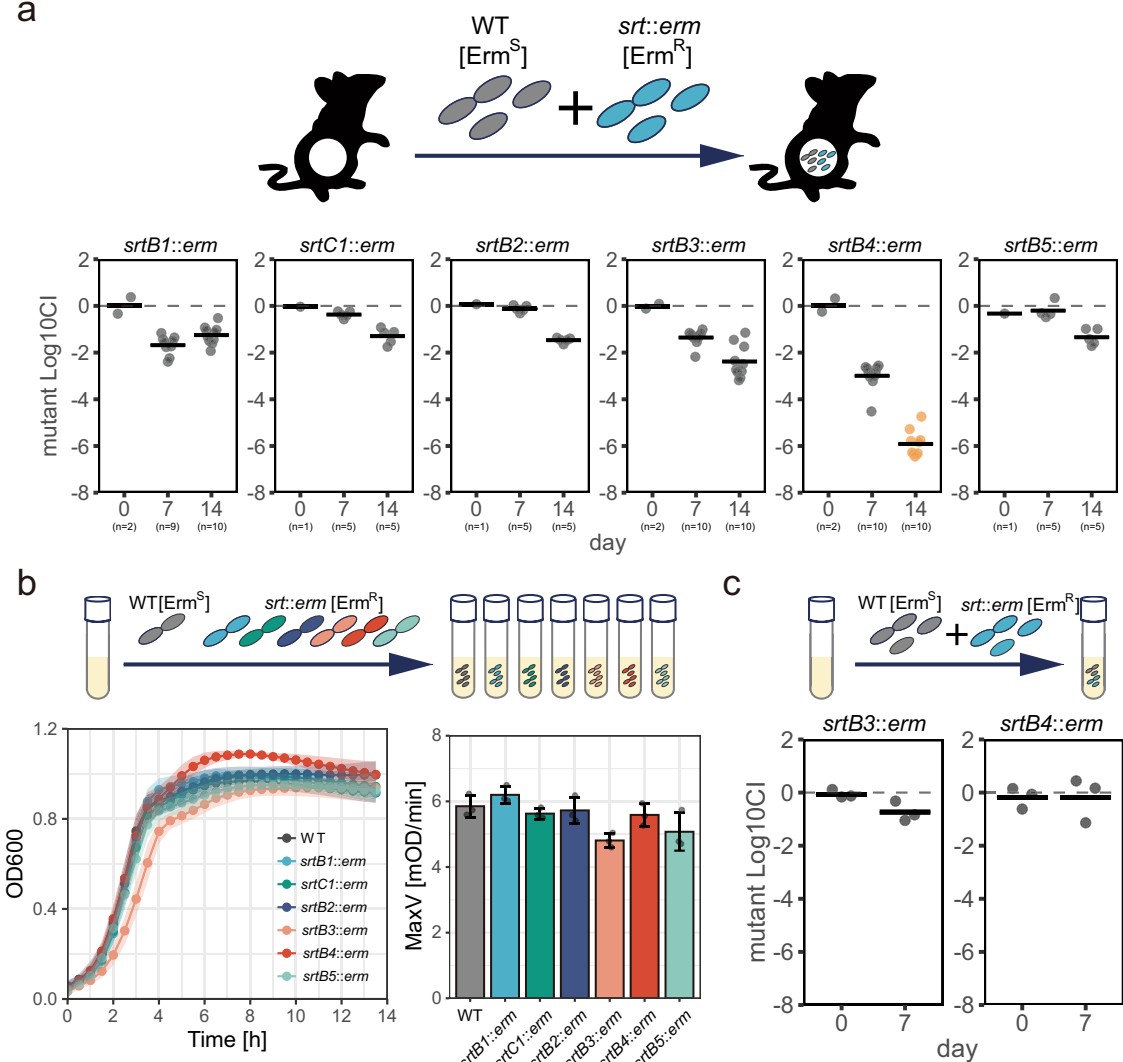

**Fig. 4 | Sortase genes are involved in the competitive fitness of *M. gnavus* during mouse intestinal colonization. a** Competitive colonization against *M. gnavus* wild-type in the intestine of germ-free mice. *M. gnavus* wt and each sortase mutant strain are mixed with equal CFUs to be orally inoculated into germ-free mice. *M. gnavus* wild-type and each sortase mutant strain are erythromycin-sensitive and resistant, respectively, which enables us to calculate the frequency of mutant strains in *M. gnavus* populations colonizing the intestine. The competitive index (CI) was calculated as the CFUs of the mutant strain divided by the CFUs of the wild type. The means of Log10 CI and the values of each mouse are shown by bars and plots. Orange plots indicate values below detection limits. The number of

biological replicates (n) is indicated at the bottom of the panel. **b** Growth curve of each mutant strain. *M. gnavus* wild-type and mutant strains were separately inoculated into liquid media, and O.D. 600 values were measured over time (bottom left). MaxV values acquired from growth curves are shown on the bottom right. Data represent means ± SD from three independent experiments. **c** Competitive colonization against *M. gnavus* wild-type in vitro. *M. gnavus* wt and sortase mutant strains (*srtB3* or *srtB4* mutants) are mixed and inoculated in liquid media. The competitive index (CI) was calculated as the CFUs of the mutant strain divided by the CFUs of the wild-type. The means of Log10 CI and the values of 3 biological replicates are shown by bars and plots.

## The *srtB4* gene cluster is predominantly lost in *M. gnavus* isolates from Crohn's disease patients

Loss of the *srtB4* gene and CPS production are disadvantageous for competitive colonization but are associated with aggravated intestinal inflammation in *M. gnavus* mono-associated mice. These findings suggest that the presence or absence of the *srtB4* gene cluster may be one of the factors contributing to *M. gnavus*-associated disease. Recently, complete genome sequences have become available for *M. gnavus* strains isolated from healthy individuals and from patients with Crohn's disease. Using this dataset of *M. gnavus*, 26 and 14 strains isolated from healthy persons and Crohn's disease patients, respectively, we investigated the distribution of the *srtB4* gene cluster (RGna_12030 to RGna_12060) (Fig. 7a). We found that the genomes of healthy person isolates tend to possess the *srtB4* gene cluster more frequently compared to Crohn's disease patient isolates (*srtB4*

frequency: 0.69 in healthy person isolates vs. 0.21 in Crohn's disease patient isolates), despite the limited sample size. To further confirm the association between *srtB4* and CPS production, we cultured the RI1[36] and NBRC 114413 strains, which possess and do not possess the *srtB4* gene cluster, respectively. We found that the RI1 strain displayed robust CPS production, whereas NBRC 114413 did not (Fig. 7b). We then generated a *srtB4* disruptant in the RI1 strain and confirmed reduced CPS presentation (Fig. 7b, c). These findings suggest that the *srtB4* gene cluster is associated with CPS production and that non-capsulated and inflammatory *M. gnavus* strains may be more frequently found in the intestines of Crohn's disease patients.

## Discussion

We developed a genetic toolkit for the human gut symbiont, *M. gnavus*, enabling targeted gene manipulation to investigate its roles in

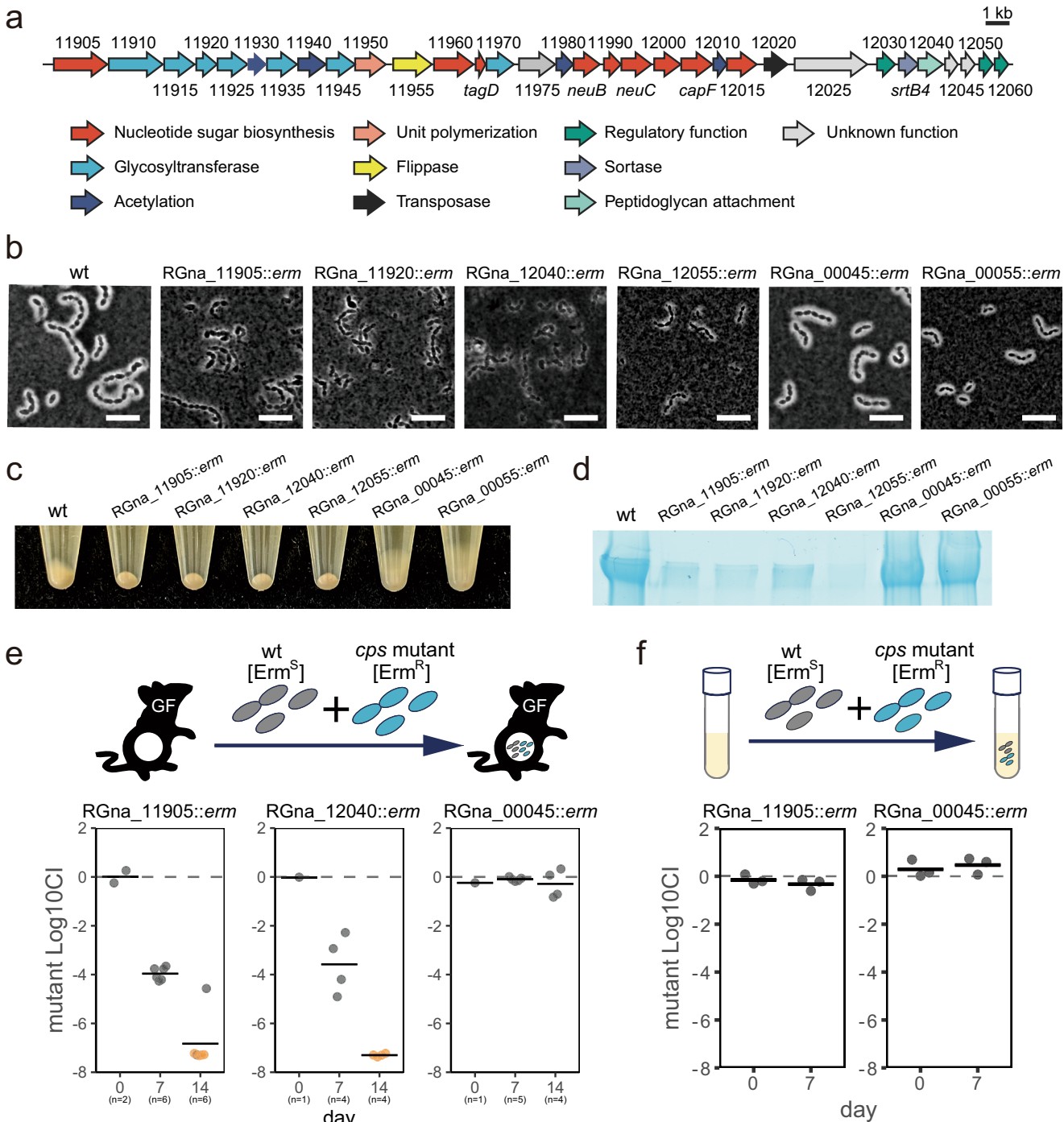

**Fig. 5 | Identification of the gene cluster mediating CPS biosynthesis in *M. gnavus*. a** A schematic of the gene cluster neighboring *srtB4*. Genes with a similar predicted function are shown in the same color. Numbers indicate the locus tag of the genes. **b** Indian ink staining of each gene mutant. Cells grown overnight were mixed with an equal volume of Indian ink and observed using a phase contrast microscope. Bar = 5 μm. **c** Cell pellets after centrifugation (10,000 × *g* for 2 min) of cell cultures are also shown. **d** Capsular polysaccharide production detected by SDS-PAGE. Capsular polysaccharide fractions collected from cells grown overnight were separated by SDS-PAGE. Capsular polysaccharides were detected by acid polysaccharide staining dye alcian blue. Each lane contains 0.5 O.D. 600 units. **e** Competitive colonization against *M. gnavus* wild-type in the intestine of germ-free

mice. *M. gnavus* wild-type and each mutant strain are mixed with equal CFUs to be orally inoculated into germ-free mice. The competitive index (CI) was calculated as the CFUs of the mutant strain divided by the CFUs of the wild-type. Each experiment used at least four mice. The means of Log10 CI and the values of each mouse are shown by bars and plots. Orange plots indicate values below detection limits. **f** Competitive colonization against *M. gnavus* wild-type in vitro. *M. gnavus* wild-type and mutant strains are mixed and inoculated in liquid media. The competitive index (CI) was calculated as the CFUs of the mutant strain divided by the CFUs of the wild-type. The means of Log10 CI and the values of 3 biological replicates are shown by bars and plots. For **b**, **d**, similar results were obtained in two independent experiments.

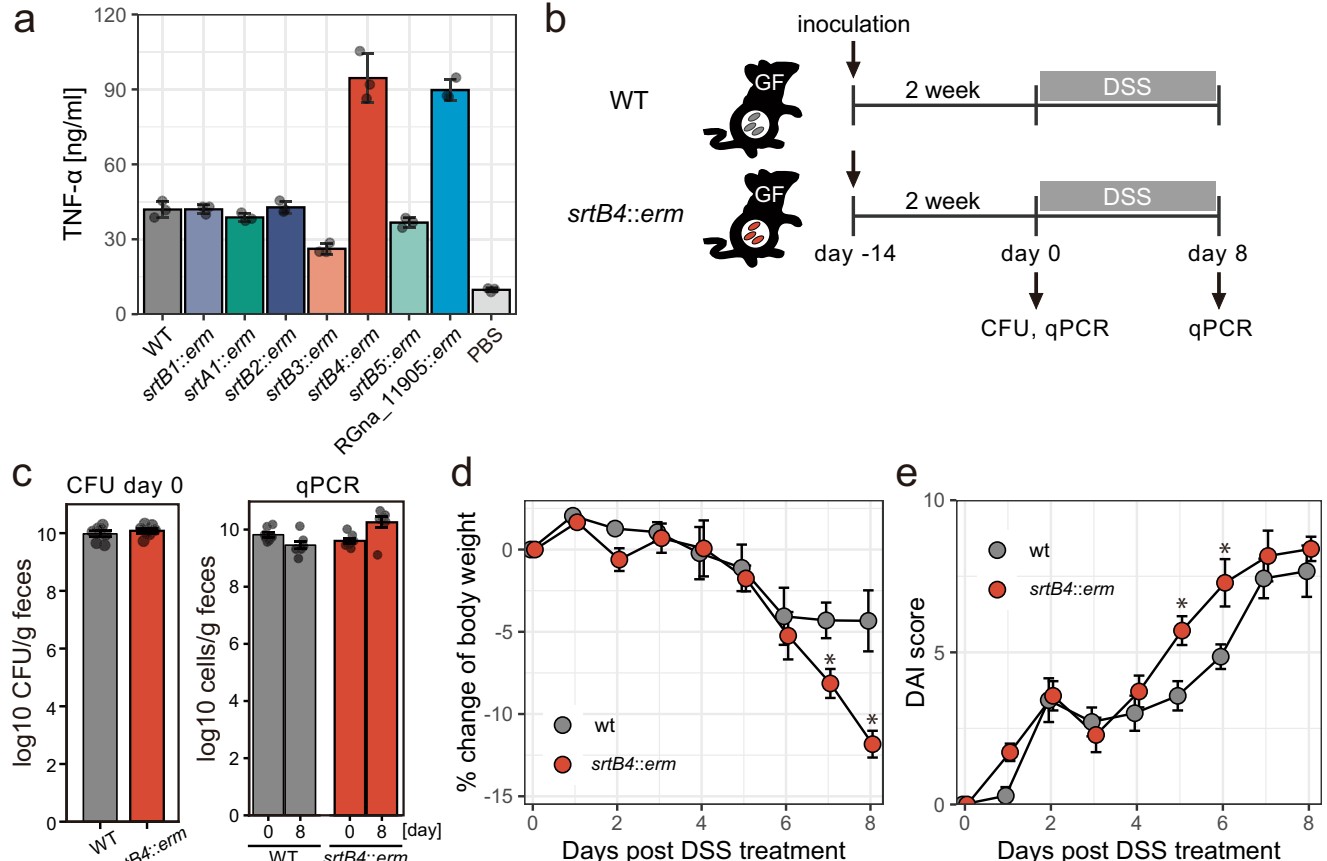

**Fig. 6 | CPS production is inversely correlated with inflammation. a** Cytokine production by mouse macrophage-like cells in the presence of *M. gnavus* cells. Mouse macrophage-like cell J774.1 ($1 \times 10^5$ cells) was incubated with *M. gnavus* wt and mutant strain ($1 \times 10^7$ cells) for 24 h. The concentrations of the inflammatory cytokine TNF-α in the culture supernatants were measured by ELISA. Bars represent means ± SD from 3 technical replicates of a representative experiment. Similar results were obtained in 3 independent experiments. **b** Schematic of the animal experiment procedure using dextran sodium sulfate (DSS). *M. gnavus* cells ($2 \times 10^7$ cells/mouse) were orally inoculated into germ-free mice ($n = 7$). **c** Quantification of *M. gnavus* cells colonizing the murine intestine. Means and standard errors of CFUs of each strain in fecal samples at day 0 are shown on the left ($n = 7$). The number of *M. gnavus* cells in feces at day 0 and 8, quantified by qPCR, is shown on the right ($n = 7$). Symptoms of gnotobiote mice during DSS administration. Means and standard errors of body weight change (**d**) and disease activity index (DAI) (**e**) are shown. Statistically significant differences calculated by two-sided Welch's t-test ($p < 0.05$) are indicated by asterisks. Exact $p$-values are 0.01593 and 0.00369 on day 7 and 8 for body weight change, and 0.00801 and 0.02172 on day 5 and 6 for DAI, respectively.

intestinal colonization and host inflammation. Despite advances in omics-based approaches that have broadened our understanding of microbiota-disease and health associations, the molecular mechanisms governing gut microbiota-host interactions remain largely elusive. A major barrier has been the lack of genetic tools for non-model gut bacteria. Recent progress in the genetic engineering of several gut bacteria, including *Akkermansia muciniphila*[37], *Fusobacterium nucleatum*[38], *Eggerthella lenta*[20], and *Segatella copri*[39] underscores the importance and growing feasibility of these approaches. Our study contributes to this expanding field by introducing a versatile genetic system for *M. gnavus*, a species strongly implicated in human diseases such as IBD.

We employed the Clostron and CRISPR-Cas systems simultaneously to construct a multiple-deletion strain. We also confirmed that the CRISPR plasmid (Tm^R) can be cured by conjugative transfer of another plasmid carrying the same origin of replication and a different antibiotic resistance marker (Erm^R). The resulting strain lost resistance to thiamphenicol (Tm^R), indicating that the original plasmid was displaced by the newly introduced one (Supplementary Fig. 8a). Theoretically, this approach could enable iterative introduction of CRISPR plasmids targeting different genes. However, we have not yet tested this strategy for multiple deletions in practice. We have attempted to use our shuttle vectors for fluorescent tagging in other gut bacterial

species closely related to *M. gnavus*. So far, we have successfully introduced plasmids into several Lachnospiraceae strains, such as *M. gnavus* isolates, *Enterocloster clostridioformis*, and *E. bolteae* (Supplementary Fig. 8b). These results suggest that our genetic manipulation strategy may be broadly applicable to other members of the Lachnospiraceae family.

A previous study suggested an inverse relationship between CPS production and the potential to induce inflammatory cytokine production by *M. gnavus* strains isolated from IBD patients[12]. A putative CPS biosynthesis gene cluster had been predicted based on the draft genome sequences of these isolates, and Nooij et al. reported that this CPS gene cluster was associated with the distinction of healthy individuals and Crohn's disease patients[19]. However, while our functional analysis using the isogenic CPS mutants supported the association between CPS and inflammation, the previously identified gene cluster did not appear to contribute to CPS biosynthesis or competitive intestinal colonization in our experimental conditions. Additionally, the *srtB4* gene, which is required for CPS presentation, appears to be less prevalent in isolates from Crohn's disease patients than in those from healthy individuals in the available complete genome dataset, although the sample size in this study is limited. This highlights a key limitation of comparative genomics alone and underscores the importance of functional validation using a bacterium-specific genetic

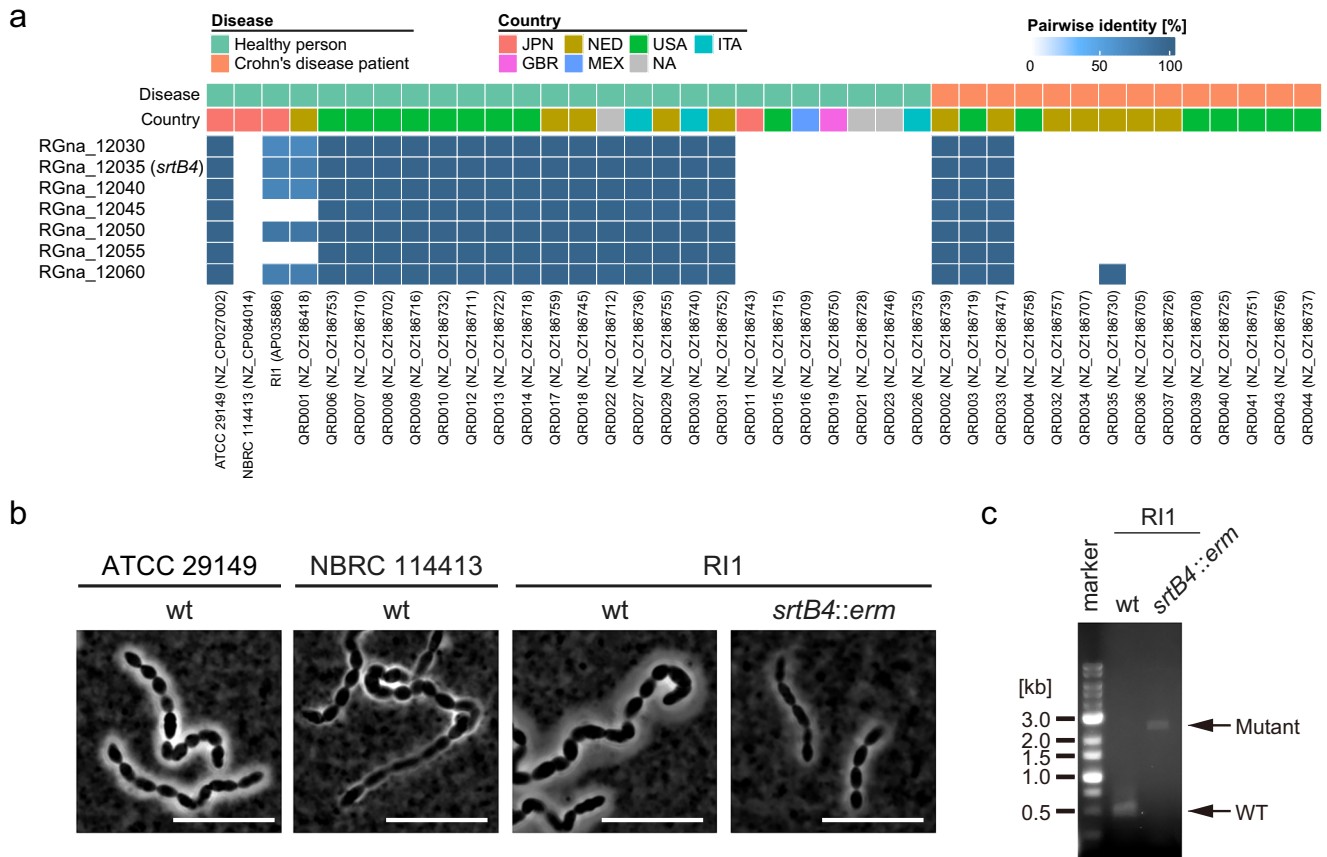

**Fig. 7 | The *srtB4* gene cluster is more frequently conserved in *M. gnavus* isolated from healthy individuals. a** The conservation and identities of the genes consisting of the *srtB4* gene cluster of the ATCC 29149 strain were compared in the genomes of isolated strains available in the public database using BLASTn. Forty complete genomes of *M. ganvus* isolates were divided into two groups, highlighted in green and red, representing healthy individuals and patients with Crohn's disease, respectively. The geography of each strain is indicated by colors. Pairwise identities (%) of each gene are shown as a heatmap. **b** CPS presentation in isolated strains. Indian ink-stained images of the *srtB4* gene cluster-negative NBRC 114413 and *srtB4* gene cluster-positive RI1, as well as the mutant strain of RI1. Bar = 10 μm. **c** Confirmation of the gene disruption by PCR targeting the *srtB4* homolog neighboring region in the RI1 strain. PCR products using the *srtB4::erm* genome as a template DNA are 2 kb larger than those using the wild-type genome.

modification system. Nevertheless, due to the limited number of isolates with complete genome sequences, the association between *srtB4* and diseases should be interpreted with caution and warrants further investigation.

Bacteria often harbor multiple loci for CPS biosynthesis clusters, which enable the production of structurally diverse CPSs and thereby facilitate adaptation to diverse environmental conditions[40]. In *B. thetaiotaomicron*, the type strain possesses eight CPS gene loci, which determine susceptibility to phages[41]. The *Lactiplantibacillus plantarum* genome contains several regions for CPS, which modulate surface glycan composition and immune responses of host cells[42]. Although the previously proposed CPS cluster may be functional under specific, yet unexplored, environmental conditions, our findings indicate that the *srtB4*-associated gene cluster is essential for CPS production in *M. gnavus* ATCC 29149. This cluster includes genes encoding LCP-family enzymes and a BY-kinase/phosphatase pair, known to catalyze the attachment of extracellular polysaccharides to peptidoglycan[34] and as regulators of CPS biosynthesis in other Gram-positive bacteria[35]. These genes are highly conserved in the *srtB4*-positive genomes (Fig. 7a). In contrast, we also found that the upstream gene region, involved in sugar biosynthesis and glycosylation, is poorly conserved (Supplementary Fig. 9a). Furthermore, comparison of the CPS locus between the type strain ATCC 29149, RI1 and NBRC 114413 strains indicates that the overall gene composition and organization vary among the strains (Supplementary Fig. 9b). Given that the upstream gene clusters mainly encode sugar biosynthesis and glycosyltransferase enzymes involved

in the determination of CPS sugar compositions, CPS composition in *M. gnavus* would be strain-dependent. Whether this variation contributes to differences in inflammatory potential among *M. gnavus* strains remains an important question for future research.

Heterogeneous expression of CPS, which is often driven by phase-variable promoters of CPS gene clusters, is a known factor that drives heterogeneity in bacterial populations, facilitating an adaptive strategy[43]. However, our promoter-reporter assays show that the promoter expression is not bimodal; it appears that all populations of the *M. gnavus* strain ATCC 29149 produce CPS. This result suggests that *M. gnavus* CPS genes are extremely low-frequency variable or not phase-variable in our experimental conditions. This contrasts with other gut bacteria and may reflect a distinct regulatory mechanism in *M. gnavus*.

Superantigens play a pivotal role in modulating host immune cells, including B cell activation and disruption of bacterial cell recognition by immune cells. *M. gnavus* possesses two copies of the superantigen genes, *ibpA* and *ibpB*, which encode cell wall-anchored proteins that bind to human IgG and IgA, thereby stimulating B cells[27]. Using the genetic toolkit as a combination of gene disruption and deletion systems, we constructed a superantigen-non-producing mutant, which may facilitate understanding of the role of superantigens in interactions with immune cells and the pathogenesis of *M. gnavus*. Moreover, we found that the *srtB3* mutant cells release the superantigens extracellularly and display a deficiency in binding to hIgG, whereas other sortase mutant cells are comparable to wild-type,

suggesting that SrtB3 specifically recognizes and catalyzes the superantigens.

In contrast, mutant strains of the other sortases (SrtB1, SrtC1, SrtB2, and SrtB5) did not exhibit significant defects in intestinal colonization under our experimental conditions. Nonetheless, these sortases may play a role in colonization or host interactions under specific environmental conditions, such as dietary variation or altered host genotypes. Additionally, we performed a phylogenetic analysis of sortase proteins derived from *M. gnavus* and various other Gram-positive bacteria (Supplementary Fig. S4a). This analysis revealed that most *M. gnavus* sortase B homologs (RGna_01825, RGna_09760, RGna_10250, RGna_12035, RGna_13420) form a distinct clade, separate from known sortase B proteins in other species. Furthermore, we found candidate target proteins encoded near these sortase genes, whose C-terminal sequences contain putative cell wall sorting signals (supplementary Fig. S4b). This suggests that they may recognize unique substrate motifs. Identifying their target substrates may facilitate understanding the functional repertoire of *M. gnavus* surface proteins that may contribute to environmental adaptation in this bacterium as a pathobiont[28].

In summary, our study provides a comprehensive genetic toolkit for *M. gnavus*, enabling the functional analysis of key genes involved in intestinal colonization and host interaction. We identify a CPS biosynthesis cluster critical for colonization and inversely linked to inflammation, and we establish the role of sortase SrtB3 in anchoring superantigens. These findings advance our understanding of how *M. gnavus* strains differ in their pathogenic potential and demonstrate the power of precise genetic tools in dissecting host-microbe dynamics. Further exploration of strain-specific factors[13] and environmental influences will be essential to fully elucidate the role of *M. gnavus* in intestinal health and disease.

## Methods

### Bacterial culture conditions
Bacterial strains and plasmids used in this study are listed in Supplementary Data 1 and 2. We used *M. gnavus* JCM 6515 (=ATCC 29149) or NBRC 114413, obtained from JCM and NBRC, respectively. We used *E. coli* HB101/pRK24 or Stellar™ for conjugation or DNA cloning. *M. gnavus* strains were grown in GAM (Shimadzu) or YCFAg (YCFA medium[44] with 0.4% glucose) media under an atmosphere of 80% $N_2$, 20% $CO_2$, and 4% $H_2$ in an anaerobic chamber (COY). *E. coli* was routinely grown aerobically in LB medium. Antibiotics were supplemented in the media when necessary; 50 µg/ml carbenicillin, 20 µg/ml chloramphenicol, 10 µg/ml Thiamphenicol, 250 µg/ml D-cycloserine, 100 µg/ml kanamycin and 200 or 10 µg/ml erythromycin for *E. coli* or *M. gnavus*.

### Plasmid construction
Oligonucleotides used in this study are listed in Supplementary Data 3. We used an indigenous *M. gnavus* NBRC 114413 plasmid to construct the *E. coli-M. gnavus* shuttle vectors. The *M. gnavus* NBRC114413 genome contains a small plasmid that should include a replicon of origin suitable for *M. gnavus*. We PCR amplified the putative *M. gnavus ori* region, including the *repA* gene, from the NBRC114413 plasmid (Supplementary Data 4). We also amplified a fragment containing an antibiotic resistance gene (*catP*), *traJ*, ColE1 *ori*, and MCS from pMTL83151, a shuttle vector used in *Clostridium* species[45]. These PCR fragments were combined by the NEB Hifi assembly system (New England Biolabs). The resultant plasmids were used for conjugative transfer to *M. gnavus* from *E. coli*. We confirmed that one of the plasmids, pNORg10, was suitable for replication in *M. gnavus* since many *M. gnavus* transconjugants could be obtained (Supplementary Fig. 1). We constructed pNORg50, in which the chloramphenicol resistance gene in pNORg10 was switched with the erythromycin resistance gene. We cloned a putative promoter region and CDS of the *srtB4* gene into pNORg50 to construct the *srtB4* complementation plasmid.

To construct a suitable expression system for *M. gnavus*, we cloned cumate-[20], lactose-[21], tetracycline-[22], and xylose-inducible promoter[23] to the upstream of mScarlet-I, red fluorescent protein gene[24], which are amplified from synthetic DNA (Invitrogen), pCPO0514[46], pRFP185, pXCH, respectively. The *cymR* and cumate operator (CuO) was artificially synthesized. We used the artificially synthesized codon-optimized fluorescent protein gene for the low GC content bacterium *C. perfringens*. For strict regulation of cumate-inducible expression, we cloned strong promoter $P_{fdx}$ into the upstream of *cymR* genes and theophylline riboswitch[47] into the SD sequence of mScarlet-I. We tested several fluorescent proteins, mVenus[48], oxStayGold[49], mNeonGreen[50], mKate2[51], for fluorescent labeling of *R. ganvus*.

For the precise deletion of target genes, we applied the CRISPR-Cas9 system. We amplified the *ncas9* (nickase) gene, sgRNA gene under the control of a synthetic constitutive promoter, and 1 kb homology arms for homologous recombination from pNICKclos2.0[52], SS9_RNA[53] and *M. gnavus* genome, respectively. pNICKclos2.0 was a gift from Sheng Yang (Addgene plasmid # 73228). SS9_RNA was a gift from Ryan Gill (Addgene plasmid # 71656). These DNA fragments are joined by HiFi DNA Assembly.

### Conjugation
The plasmids were introduced into *M. gnavus* via conjugation as described with some modifications[26]. Briefly, *E. coli* HB101/pRK24 harboring the desired plasmid was cultured overnight in 2 ml LB medium at 30 °C with vigorous shaking. *E. coli* cells were collected from 1 ml culture by centrifugation (1500 × *g*, 3 min), washed with PBS, and placed into an anaerobic chamber. The *E. coli* cell pellet was suspended in 200 µL of *M. gnavus* culture grown in YCFAg medium overnight at 37 °C. We dropped the mixture (10 × 20 µL) onto the GAM plate and incubated it for 6 h at 37 °C. The transconjugants on the plate were collected in 1 ml PBS and spread onto GAM-TCK (Thiamphenicol-D-cycloserine-kanamycin) or -ECK (Erythromycin-D-cycloserine-kanamycin) plates. We typically observed the colonies of the transconjugants after 1–2 days of incubation at 37 °C.

To construct *erm*-inserted mutant strains, the transconjugants were cultivated in YCFAg liquid culture containing thiamphenicol, which was then spread onto GAM plates containing erythromycin. We confirmed intron-*erm* gene insertion in the appropriate locus by PCR.

For the construction of deletion mutants using the CRISPR-Cas9 system, we picked the colonies of transconjugants and re-streaked them onto GAM containing thiamphenicol or erythromycin. We checked the deletion of the target locus by PCR.

### Fluorescent reporter assay
*M. gnavus* strains harboring plasmids expressing fluorescent proteins were grown anaerobically in YCFAg media containing the appropriate antibiotics. After culture, we removed the culture from the anaerobic chamber and washed the cells with PBS. The cell suspensions were agitated by using a ThermoMixer (Eppendorf) at 1400 rpm to be exposed to oxygen for 2 h at room temperature or overnight at 4 °C to prompt the maturation of fluorescent proteins. The fluorescent values were measured using a microplate reader Synergy H1 (BioTek).

### Western blotting
*M. gnavus* cells were pelleted by centrifugation at 20,000 × *g* for 2 min. The supernatant was mixed with 2 volumes of ethanol. Then, the extracellular proteins were collected by centrifugation at 20,000 × *g* for 5 min. The cell pellets were washed with PBS and suspended in protoplast buffer (50 mM Tris-HCl pH 7, 50 mM $MgCl_2$, and 20% (w/v) sucrose) containing 25 mg/ml lysozyme (Wako 129-06723) and 40 U/ml mutanolysin (Merck M9901)[54]. The suspensions were incubated at 37 °C and 1400 min$^{-1}$ for 2 h and then centrifuged at 20,000 × *g* for 2 min. Microscopic observation confirmed that all cells were converted

to protoplasts after the treatment. We used the supernatant as a cell wall fraction and the pellets as a cytoplasmic fraction. All samples were suspended in SDS-sample buffer (0.125 mol/L Tris-HCl, pH 6.8, 4% SDS, 20% Glycerol, 0.01% BPB, 10% 2-mercaptoethanol) and heated at 95 °C for 10 min. The protein samples were separated by SDS-PAGE and electroblotted onto PVDF membranes. The membranes were blocked with Bullet Blocking One (Nacalai Tesque). Superantigen proteins were probed with human IgA (Bethyl Laboratories P80-102) diluted 1:5000 with PSB containing 0.05% Tween 20. These antibodies were labeled with anti-human IgA-HRP (Invitrogen A18787) diluted 1:20000. The bound antibodies were labeled with Immunostar LD (Wako) and detected using a C-DiGit System (LI-COR Biosciences).

## Flowcytometry
Cells grown for 6 h were adjusted to a concentration of $1 \times 10^7$ CFU/ml with PBS. After centrifugation, the cells were resuspended in 2% BSA/PBS and incubated at room temperature for 15 min. The cells were then probed with 5 µg/mL of mouse IgG Alexa Fluor 488 conjugates (Thermo Fisher Scientific, A-10631; lot number, 2400911) in 2% BSA/PBS and incubated for 30 min at 4 °C. After staining, the cells were washed twice with PBS and analyzed by flow cytometry using a FACS Aria II (BD Biosciences). Cell-bound IgG was evaluated based on all events detected by flow cytometry. Bacterial cell populations were gated by FSC-A/SSC-A. Fluorescence was excited with a 488-nm laser, and emission was subsequently collected using a 525/50-nm filter. Data were analyzed using FlowJo V10.8. Representative examples of the gating strategies used are provided in Supplementary Fig. 10.

## Gnotobiotic mouse experiment
The University of Tsukuba Animal Experiment Committee approved all germ-free mouse experiments. Mice were housed in a room maintained at 23.5 ± 2.5 °C and 52.5 ± 12.5% relative humidity under a 14:10-h light: dark cycle. We cultured *M. gnavus* overnight in YCFAg at 37 °C and collected the cell pellets by centrifugation in the anaerobic chamber. The cells were resuspended in GAM medium at a concentration of $1 \times 10^8$ CFU/ml. We orally gavaged 200 µL suspension into female BALB/cCr germ-free mice (6–12 weeks old). We collected fecal samples weekly and spread the fecal suspensions onto GAM with or without 10 µg/ml erythromycin to measure the CFU of wt and mutant strains in the fecal samples. We calculated competitive fitness (CI) by dividing the CFU of the mutant strain by that of the wild type.

To assess whether *M. gnavus* mutant exacerbates intestinal inflammation, we administered 2.5% dextran sodium sulfate (MP Biomedicalsas 160110) as drinking water to *M. gnavus* mono-associated male BALB/cCr mice (8-10 weeks old) and daily monitored body weight changes and disease activity scores (DAI). The DAI scores were evaluated on a scale of 0–4 for the following observed parameters weight loss, stool consistency, and bleeding[55]. Weight loss was scored as follows: 0 for no weight loss, 1 for 0–5% weight loss, 2 for 5–10% weight loss, 3 for 10–15% weight loss, and 4 for more than 15% weight loss. Stool consistency was scored as 0 for normal stool, 2 for soft stool, and 4 for diarrhea. Bleeding was scored as 0 for no bleeding, 2 for slight bleeding, and 4 for severe bleeding. Total scores were used to indicate intestinal inflammation.

Sex as a biological variable was not specifically analyzed because only one sex was used for each assay (female mice for colonization, male mice for DSS).

## In vitro competition assay
The overnight cultures of wt and mutant strains were diluted 1:50 in 200 µL of YCFAg medium in a 96-well plate and anaerobically grown at 37 °C. We sub-cultured the cells every 24 h. On day 7, we diluted the culture daily in fresh YCFAg and spread it on the GAM plate to count the CFUs of each strain.

## CPS analysis
We mixed *M. gnavus* culture with Indian ink at an equal volume and then observed the cells with a phase contrast microscope.

To detect acid polysaccharide production, we anaerobically cultured *M. gnavus* cells in YCFAg at 37 °C for 16 h and collected cell pellets by centrifugation at $20,000 \times g$ for 2 min. The cell pellets were washed with PBS and resuspended in $H_2O$. The suspension was heated at 95 °C for 10 min to collect cell-bound CPS. We collected the suspension supernatant after centrifugation and used it for SDS-PAGE. Samples corresponding to 0.5 $O.D._{600 \text{ nm}}$ unit were loaded into each lane. Extracted CPS was visualized by the cationic dye alcian blue (0.125% alcian blue, 40% ethanol, and 5% acetic acid) for 2 h and washed with 40% ethanol and 5% acetic acid for 2 h[56].

Transmission electron microscopy was outsourced to Hanaichi UltraStructure Research Institute (Aichi, Japan). *M. gnavus* cells were fixed in 2.5% glutaraldehyde/1.25% paraformaldehyde/0.03% picric acid with 10 mM lysine and 0.1% ruthenium red in 0.1 M sodium cacodylate buffer (pH 7.4).

## Cytokine ELISA
J774.1, a mouse macrophage-like cell line, purchased from RIKEN BRC, was cultured in RPMI1640 medium supplemented with 10% fetal bovine serum at 37 °C under 5% $CO_2$ atmosphere. *M. gnavus* grown in YCFAg overnight was suspended in PBS to reach $O.D._{600 \text{ nm}} = 10$ (approximately $1 \times 10^9$ cells/ml). One mL of J774.1 cell suspension at a $1 \times 10^5$ cells/ml concentration was incubated with 10 µL of *M. gnavus* cells (MOI = 1: 100) for 24 h. The cell culture supernatant was collected to detect cytokine production by the Murine TNF-α ELISA Development Kit (PeproTech 900-K54K) according to the manufacturer's instructions.

## Bacterial quantification by qPCR
Bacterial DNA was extracted from fecal samples by a bead-beating method. Freeze-dried feces (10 mg) were suspended in 400 µL of 1% SDS/TE buffer and mixed with 400 µL glass beads (Merck G8772) and an equal volume of phenol-chloroform-isoamyl alcohol (Nippon Gene). Bead beating was carried out using Shake Master Neo at 1400 rpm for 15 min. After centrifugation at $20,000 \times g$ for 5 min, supernatants were collected for further purification of nucleic acids using Monarch® Spin gDNA Extraction Kit (New England Biolabs) according to the manufacturer's instructions. DNA concentration was quantified using Quant-iT™ PicoGreen™ dsDNA Assay Kits (Thermo Fisher Scientific).

Bacterial quantification by qPCR was performed as described previously[57]. The 16S rRNA gene fragments were amplified by PCR and diluted to 1.67 ng/µL, corresponding to $10^9$ copies/µL. The DNA solutions serially diluted to $10^8$ to $10^3$ copies/µL were used as the standard. The qPCR was carried out using TB Green® Premix Ex Taq™ II FAST qPCR (TAKARA bio) and StepOnePlus™ Real-Time PCR Systems (Thermo Fisher Scientific) according to the manufacturer's instructions. The number of bacterial cells/g feces was calculated using the Ct values, the standard curve and the amount of feces used for DNA extraction.

## Genome analysis
*M. gnavus* complete genome information was obtained from NCBI. The BLASTn analysis against srtB4 and its adjacent genes in ATCC 29149, the reference sequence, was performed using Geneious Prime 2025.0.3. (https://www.geneious.com).

## Statistics and reproducibility
Data are represented as mean values ± SD (standard deviation) or ±SEM (standard error of the mean), calculated using ggplot package in R. Significance was calculated by an unpaired Welch's t-test with Bonferroni correction for multiple comparisons using Microsoft Excel

for Microsoft 365. No statistical method was used to predetermine sample size. No data were excluded from the analyses. The experiments were not randomized. The Investigators were not blinded to allocation during experiments and outcome assessment.

## Reporting summary

Further information on research design is available in the Nature Portfolio Reporting Summary linked to this article.

## Data availability

The data supporting the findings of this study are available within this article and the Supporting Information. The source data are provided with this paper as a Source data file. Plasmids are available from Addgene (Plasmid ID, 249340-249345). Whole genome sequence data generated in this study have been deposited in the DDBJ Sequence Read Archive (DRA) under accession numbers DRR899491-DRR899492. Additional details on the datasets of this study are available from the corresponding author on reasonable request. Source data are provided with this paper.

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

## Acknowledgements

This work is supported by the Japan Society for the Promotion of Science (JSPS) KAKENHI Grant Number 25K01926 (N.O.), Institute for Fermentation, Osaka (IFO) (N.O.), JSPS KAKENHI Grant Number 23H05471 (N.N.), Japan Science and Technology Agency (JST) ERATO Grant Number JPMJER1902 (S.F.), Food Science Institute Foundation (S.F.). We wish to thank Yu Obana and Tatsuji Takahashi for their technical assistance.

## Author contributions

Conceptualization, N.O. and S.F.; methodology, N.O. and G.N.; investigation, N.O.; formal analysis, N.O.; writing—original draft, N.O.; writing—review & editing, all authors; supervision, N.N., and S.F.; funding acquisition: N.O., N.N., and S.F.

## Competing interests

The authors declare no competing interests.
