## [Transparent Peer Review file · Nature Communications]

A genetic toolkit for the human gut bacterium *Mediterraneibacter gnavus* identifies capsular polysaccharides as a competitive colonization factor

Corresponding Author: Professor Shinji Fukuda

Version 0:

Reviewer comments:

Reviewer #1

(Remarks to the Author)

This study reports the development of a genetic toolkit for the gut bacterium *Ruminococcus gnavus*. The authors established a shuttle vector, inducible expression, fluorescent protein reporter, and two gene deletion methods for this organism. Using the genetic deletion approach, they generated mutants with individual deletions of putative sortase-encoding genes, and found that *srtB3* is responsible for anchoring a bacterial superantigen to the cell wall, and *srtB4* is essential for capsular polysaccharide production and bacterial *in vivo* fitness. This work will be impactful for the future mechanistic studies of bacterial physiology and host-microbe interactions in *R. gnavus* and potentially other closely related bacteria.

Major:

Genetic toolkit:

For Line 103 and Fig. 1D-1F, please include genotyping results from PCR and/or sequencing in the supplementary figures. Regarding the strategy to create the *ibpA/B* double mutant, why did authors use a clostron-based method to delete *ibpA* and then use CRISPR/Cas to remove *ibpB*?

From Fig. 1E, it appears that only the promoter region and upstream portion of the *ibpB* CDS were deleted rather than a full deletion. Have the authors attempted to delete the entire *ibpB* CDS or even the *ibpA/ibpB* region together?

Is it possible to cure the CRISPR plasmid and perform iterative gene deletions in *Ruminococcus gnavus*?

How successful is the genetic manipulation strategy in other *Ruminococcus* strains?

Sortase deletion:

For the duplicated *srtA2* and *srtB6* genes, would it be possible to use a different antibiotic resistance marker after the first deletion to disrupt the second copy?

How different are these sortases in terms of protein sequences and genomic context? Any predictions regarding their substrate preferences? An illustration in the SI would aid in inferring their biological roles.

SrtB3 function:

Control proteins should be included in the WB to demonstrate clear separation of different bacterial fractions.

SrtB4 function:

The paper does not explain how *srtB4* is involved in CPS biosynthesis. Have the authors tested which proteins are the actual substrates of *SrtB4*?

Fig. 6: What are the colonization levels of the two strains in the mono-colonization experiments? Could the observed differences in colitis severity be explained by differences in colonization levels?

Fig. 7: The comparison of three strains is not strong evidence to link *srtB4* to CPS production. Including additional strains or demonstrating gene deletion in the R11 strain would strengthen the conclusion.

Minor:

Line 52: The description linking *R. gnavus* to IBD severity is vague. Please clarify whether it is the abundance or presence of *R. gnavus* that is associated with symptoms, and whether this is a positive correlation.

Line 93: please explain the term "SD" sequence.

Fig. 1F, 2A, and others: Please indicate the expected location of the targeted proteins in the WB.
The methods section should describe how complementation experiments corresponding to Fig. 3F and 3G were performed. These complementation strains are not listed in Table S1.
Please consider depositing the plasmids to Addgene to facilitate distribution. Including the replicon sequence in the SI table would also be helpful.
Line 288: Please provide more detail regarding strain dependence. In Supplementary Figure 6, many strains appear to lack the upstream genes involved in sugar biosynthesis and glycosylation. In those strains, which genes are present instead? Are they homologs of the genes in ATCC 29149, or are they entirely different genes?

Reviewer #2

(Remarks to the Author)

This study begins with the construction of a suite of genetic tools that greatly enhance the genetic systems currently available to work with *M. gnavus*. These include an inducible gene expression system where two inducers lead to reduced leaky expression. The team also designs constructs for gene deletions and constructs for fluorescent protein analysis in *M. gnavus*. The bulk of the study addresses the role of various sortases in the linking of CPS and proteins to the cell using isogenic mutants. Most of these studies are well performed and provide new important data. The study includes the use of gnotobiotic mice to show fitness defects of particular mutants during competitive colonization with WT. The studies included in Figure 6 and 7 are the weakest of the study and relate to inflammatory responses or protection due to the CPS, and genetic analyses of the CPS locus in healthy and Crohn's disease patients. In total, this study significantly contributes to the field, but some areas need additional experimentation or analyses to better justify the conclusions.

The genetic tools are a significant advance and must be made available through AddGene or another not for profit plasmid distribution company.

The newly reclassified genus must be used in the title and throughout and initially the authors should write "formerly *Ruminococcus gnavus*".

Lines 27 and 28 of the abstract should be rewritten to state that "mutations of six of the eight sortase encoding genes allowed for the identification of which are involved in the surface localization of the CPS and proteins predicted to have superantigen properties." As written, it sounds as if you show that sortases are involved in surface attachment of molecules in Gram positive bacteria, which is well established.

To say that the organism is a "commensal" and then say that it is associated with many diseases is not correct terminology. The term symbiont should be used or you can say it if frequently a member of the human gut microbiota.

The caveat that these aero-intolerant bacteria must be exposed to oxygen for two hours for fluorophore maturation should be discussed. It is assumed that cultures were shaken during this exposure, which if so, should be stated in the methods.

Line 53 "This implies that *R. gnavus* could induce inflammation by blooming in the intestines." This does not make that implication. They may just be able to better survive in inflammatory states.

Line 60 – change "disease activities" to "inflammation"

Line 71 "specific sortase enzymes"

The last sentence of your introduction is another obvious statement and could be better tailored to your findings

Line 335 – remove the word "sequence". Give a brief description of pMTL83151 "a shuttle plasmid used in *Clostridiodes* species".

Line 108 – what is meant by two copies of the gene, *ibpA*, and *ibpB*? Do you mean one copy of each? If so, just write encoded by genes *ibpA*, and *ibpB*.

Line 110 – Did you do a whole genome sequence of this strain? This is now quite inexpensive and is necessary to show that there are not off-target effects from the CRISPR/Cas system.

Line 113- better to write "other members of the Lachnospiraceae family".

Line 121 – give your rationale for naming two of the sortase genes *srtA* and 6 of them *srtB*.

Line 130 – maybe better to say "in sortase-processed cell wall anchored proteins"

Figure S3C –As you have a genome sequence, did it reveal two copies of this gene cluster? This should be performed with long read technology to overcome problems assembling repetitive regions. Is this region likely a mobile genetic element? If so, these *srt* genes may not be present in present in all *M. gnavus* strains. MGEs often insert into tRNA genes, which you could search for at the junctions of the duplicated region. What other genes are present in this duplicated region?

Line 148 – better to say “formed a less compact pellet”

Fig 3B and 3F must show a MW range so that we can determine the size of the CPS. They are typically quite heterogeneously sized high MW molecules.

Fig 3C – are there any other phenotypes of the *srtB4* mutant? In the image, it appears to have shorter chains. Do these bacteria clump at all as is a characteristic of some bacteria when their capsules are removed.

Line 152-154 – this image is not enough proof to say that these are fimbriae. These bacteria were processed for EM which introduces all types of artifacts. This statement about fimbriae should be removed.

The figure legend of Fig 3 (and line 161) should be modified, *srtB4* is not responsible for capsular polysaccharide production, it is likely important in linking the CPS to the bacterial surface. Did you analyze the supernatants for the presence of CPS? An antibody to this molecule would substantially help with analyses – which should be easily obtained by adsorbing antibodies to the WT bacterium with the CPS mutant.

Line 167 –state “competitive advantage of the WT strains against the sortase mutants”, rather than the other way around.

Lines 200-201 – This statement is not supported as written. Bacteria frequently make more than one CPS/EPS. It is quite likely that the locus identified in the other study also produces CPS that is not the same as the CPS of your study. Many CPS do not extend so far from the cell and are not nearly as visible with India ink staining as the CPS you identified. This statement should be qualified to say “these gene disruptions did not affect the phenotypes tested in our study”. In fact, the cell pellets of these deletion mutants seem to be even more loosely packed than the WT strain, suggesting some role in production or presentation of surface PSs.

Line 204 – there is no evidence that *SrtB4* affects CPS production, just likely its surface attachment, or the attachment of another protein that is necessary for its surface localization.

There are some concerns with the CPS mutants analyzed for competitive colonization in the mouse gut. As gene 11905 encodes a nucleotide-sugar synthesis gene, it may affect the production of nucleotide linked sugars necessary for other cell glycans. However, there are also concerns with any mutants that allow the incomplete building or flipping and polymerization of the repeat unit, as these mutants sequester and prevent recycling of undecaprenol-P, which is necessary for PG synthesis and can contribute to growth defects. The best strategy is to delete all glycosyltransferase genes so that the repeat unit is not built at all. These caveats should at least be discussed for the competitive colonization assays, especially since the strains have an *in vitro* growth defect. However, the 12040 mutant, which based on gene proximity may only be involved in attaching the CPS (although that is unknown), also confirms the competitive defect.

Lines 215-216 – “The CPS produced by the gene products of the identified locus....”. There may be more than one CPS/EPS produced by this organism.

Line 220 – this statement is too broad, and the opposite has also been shown. Maybe say “to have various roles in immune recognition or modulation”.

Fig 6 – there is some concern for the very different amounts of TNF in the strains comparing panels A and B. Why is there such a difference in the WT and *srtB::erm* strains in these panels?

Figure 6C – it is not evident when these mice were gavaged, please put an arrow on the time line. Also, it is surprising that mice monocolonized with *M. gnavus* survived for 8 days on 2.5% DSS.

Fig 6D – for this experiment, you must quantify the cfu/gram of bacteria in these mice over the course of the experiment (for a few time points). It is very possible that the mutant strain was not able to colonize the DSS treated mouse to the same extent as the WT, which would be relevant data. Based on the data shown, I disagree with the statement “These results suggest that CPS production inhibits the recognition of *R. gnavus* cells from host immune cells and that CPS233 deficient *R. gnavus* has the potential to induce more severe inflammation during intestinal colonization”. The inflammation is induced by the DSS, not the *M. gnavus*. Without cfu/g, there cannot be a statement about effects of the CPS in this assay.

Studies related to figure 7 are problematic

- 1) are you sure that all of these strains were isolated from separate people and are not replicates (possible longitudinal samples) from the same individual? In addition, the geographic location of the isolates (from where they live) is necessary as there can be clustering of isolates in different countries which would confound the data. There is some concern for the low number of samples for this analysis, which may be lower if duplicate samples from the same person are revealed.
- 2) CPS loci are often heterogeneous across strains of a species. Do the strains that do not have the same CPS locus as described in this study have a distinct CPS locus in this region? In Fig S6, please show gene maps of this region in other strains.
- 3) it is quite possible that another Srt can serve the function of *SrtB4* if there are diverse CPS loci in this region.

Statements in the discussion related to findings of Crohn’s disease patients are overstated based on the caveats mentioned above.

Version 1:

Reviewer comments:

Reviewer #1

(Remarks to the Author)

The revised manuscript shows improved quality, and I now have only a minor comment.

At Lines 120 and 285–295, the associated data (genomic analysis, plasmid loss, and transformation of other species) should be included in the supplementary materials.

Reviewer #2

(Remarks to the Author)

The authors did a good job of addressing my comments with addition of new data and text alterations. I have no further comments.

REVIEWER COMMENTS

Reviewer #1 (Remarks to the Author):

This study reports the development of a genetic toolkit for the gut bacterium Ruminococcus gnavus. The authors established a shuttle vector, inducible expression, fluorescent protein reporter, and two gene deletion methods for this organism. Using the genetic deletion approach, they generated mutants with individual deletions of putative sortase-encoding genes, and found that srtB3 is responsible for anchoring a bacterial superantigen to the cell wall, and srtB4 is essential for capsular polysaccharide production and bacterial in vivo fitness. This work will be impactful for the future mechanistic studies of bacterial physiology and host-microbe interactions in R. gnavus and potentially other closely related bacteria.

Response: We sincerely appreciate the reviewer's positive evaluation. In response to your suggestions, we have carefully revised the manuscript as the reviewer suggested and have addressed each comment point-by-point. We hope the revisions improve the manuscript and meet your expectations.

Major:

Genetic toolkit:

For Line 103 and Fig. 1D-1F, please include genotyping results from PCR and/or sequencing in the supplementary figures.

Response: In the revised manuscript, we have added PCR-based genotyping results to Supplementary Fig. S3. The insertion of intron-*ermB* into the *ibpA* locus resulted in a larger amplicon when PCR was performed with *ibpA*-specific primers, confirming the successful insertion of the intron. As for *ibpB*, deletion of the promoter region resulted in a smaller amplicon in PCR with *ibpB*-specific primers. In addition, when using primers targeting the internal region of the deleted sequence, we did not obtain any PCR amplicon, consistent with the expected deletion. Despite the PCR amplicon was detected in the $\Delta ibpB$ single mutant, primers annealing to the *ibpA* region, which possesses a sequence similar to *ibpB*, likely amplify the PCR product. These results confirm the successful disruption of both *ibpA* and *ibpB* genes.

*Regarding the strategy to create the *ibpA/B* double mutant, why did authors use a clostron-based method to delete *ibpA* and then use CRISPR/Cas to remove *ibpB*?*

Response: We initially attempted to construct each *ibpA* and *ibpB* mutant using the Clostron system. However, the *ibpB* disruptant retained a truncated version of the *IbpB* protein, likely due to the position of the intron insertion site, which was located near the C-terminal region of the *ibpB* CDS. We confirmed the truncated *IbpB* expression by Western blotting with human IgA, indicating that the truncated *IbpB* retained functionality. Therefore, we employed the CRISPR/Cas system to generate a loss-of-function mutation in *ibpB*.

*From Fig. 1E, it appears that only the promoter region and upstream portion of the *ibpB* CDS were deleted rather than a full deletion. Have the authors attempted to delete the entire *ibpB* CDS or even the *ibpA/ibpB* region together?*

Response: At first, we attempted to generate a complete deletion of the *ibpB* gene by homologous recombination, but this approach was not successful despite several attempts. We have not attempted to delete the entire *ibpA/ibpB* region.

Given that shorter deletions are generally more feasible with homologous recombination, we instead attempted to delete the promoter region and the upstream portion of the *ibpB* CDS.

*Is it possible to cure the CRISPR plasmid and perform iterative gene deletions in *Ruminococcus gnavus*?*

Response: Yes, we have confirmed that the CRISPR plasmid (Tm^R) can be cured by conjugative transfer of another plasmid carrying the same origin of replication and a different antibiotic resistance marker (Erm^R). The resulting strain lost resistance to thiamphenicol (Tm^R), indicating that the original plasmid is displaced by the newly introduced one. Theoretically, this approach could enable the iterative introduction of CRISPR plasmids targeting different genes. However, we have not yet tested this strategy for multiple deletions in practice. We appreciate the reviewer's suggestion and agree that iterative deletions would be a valuable application for future studies. We added the above description to the revised manuscript.

L285. "We employed the Clostron and CRISPR-Cas systems simultaneously to construct a multiple-deletion strain. We also confirmed that the CRISPR plasmid (Tm^R) can be cured by conjugative transfer of another plasmid carrying the same origin of replication and a different antibiotic resistance marker (Erm^R). The resulting strain lost resistance to thiamphenicol (Tm^R), indicating that the original plasmid is displaced by the newly introduced one. Theoretically, this approach could enable iterative introduction of CRISPR plasmids targeting different genes. However, we have not yet tested this strategy for multiple deletions in practice. "

*How successful is the genetic manipulation strategy in other *Ruminococcus* strains?*

Response: We have attempted to use our shuttle vectors for fluorescent tagging in other gut bacterial species closely related to *R. gnavus*. So far, we have successfully introduced plasmids into several Lachnospiraceae strains, such as *R. gnavus* isolates other than the type strain, *Enterocloster clostridioformis*, and *E. bolteae*. These results suggest that our genetic manipulation strategy may be broadly applicable to other members of the Lachnospiraceae family. We added the above description to the revised manuscript.

L291. "We have attempted to use our shuttle vectors for fluorescent tagging in other gut bacterial species closely related to *M. gnavus*. So far, we have successfully introduced plasmids into several Lachnospiraceae strains, such as *M. gnavus* isolates, *Enterocloster clostridioformis*, and *E. bolteae*.

These results suggest that our genetic manipulation strategy may be broadly applicable to other members of the Lachnospiraceae family.”

Sortase deletion:

For the duplicated srtA2 and srtB6 genes, would it be possible to use a different antibiotic resistance marker after the first deletion to disrupt the second copy?

Response: In the Clostron system, the *ermB* gene is initially disrupted by the insertion of the group II intron and becomes functional after the intron is spliced out upon successful retrohoming into the genome. To our knowledge, only *ermB* and *kanR* have been practically adopted as selective markers for the Clostron system. However, *R. gnavus* exhibits intrinsic resistance to kanamycin, limiting the availability of selection markers. Therefore, we did not attempt to use the intron for multiple gene disruption in this study.

How different are these sortases in terms of protein sequences and genomic context? Any predictions regarding their substrate preferences? An illustration in the SI would aid in inferring their biological roles.

Response: We appreciate the reviewer's insightful suggestion. To address this point, we have added a phylogenetic analysis of sortase proteins derived from *R. gnavus* and various other Gram-positive bacteria (supplementary Fig. 4A). This analysis revealed that most *R. gnavus* sortase B homologs (RGna_01825, RGna_09760, RGna_10250, RGna_12035, RGna_13420) form a distinct clade, separate from known sortase B proteins in other species. This suggests that they may recognize unique substrate motifs. Additionally, we found that RGna_09235 and RGna_13410, which were annotated initially as SrtA, were clustered with the sortase C proteins in the phylogenetic tree. Based on this observation, we have renamed these genes as *srtC1* and *srtC2*, respectively.

In addition, we checked genes encoded near these sortase and found candidate targets, which have putative cell wall sorting signals (supplementary Fig. 4B). This suggest that each sortase recognizes specific cell wall sorting signals. We agree that these results suggest potential functional divergence among sortase paralogs and have incorporated this interpretation into the revised manuscript.

L348. “Additionally, we performed a phylogenetic analysis of sortase proteins derived from *M. gnavus* and various other Gram-positive bacteria (supplementary Fig. S4A). This analysis revealed that most *M. gnavus* sortase B homologs (RGna_01825, RGna_09760, RGna_10250, RGna_12035, RGna_13420) form a distinct clade, separate from known sortase B proteins in other species. Furthermore, we found candidate target proteins encoded near these sortase genes, whose C-terminal sequences contain putative cell wall sorting signals (supplementary Fig. S4B). This suggests that they may recognize unique substrate motifs. Identifying their target substrates may facilitate understanding the functional repertoire of *M. gnavus* surface proteins that may contribute to environmental adaptation in this bacterium as a pathobiont²⁸.”

SrtB3 function:

Control proteins should be included in the WB to demonstrate clear separation of different bacterial fractions.

Response: We appreciate the reviewer's suggestion. However, due to the limited functional and mechanistic characterization of *R. gnavus* to date, there are very few proteins with well-established subcellular localization in this organism. Furthermore, commercially available antibodies specific to *R. gnavus* proteins are not currently accessible. This makes it challenging to include canonical controls for protein fractionation (e.g., cell wall fraction) in our Western blotting. We also note that enzymatic reactions with lysozyme and mutanolysin were used to extract cell wall fractions. Microscopic observation confirmed the formation of protoplast-like cells (see below). And we added the description in the Materials and Methods section of the revised manuscript as follows:

L438. "Microscopic observation confirmed that all cells were converted to protoplasts after the treatment."

Supporting information 1. Phase contrast microscopy of *R. gnavus* cells after the treatment of lysozyme and mutanolysin.

Furthermore, to address this limitation, we complemented our analysis with flow cytometry using human IgG, which binds to lbp proteins on the bacterial surface. Because flow cytometry detects surface-bound proteins on intact cells without cell lysis, it serves as an independent validation of surface localization. The flow cytometry results are consistent with our Western blot results, demonstrating that *srtB3* is required for surface localization of lbp proteins.

SrtB4 function:

The paper does not explain how srtB4 is involved in CPS biosynthesis. Have the authors tested which proteins are the actual substrates of SrtB4?

Response: We appreciate the reviewer's suggestion. We identified the RGna_12050 gene, located downstream of the *srtB4* gene, as a candidate substrate because it is highly conserved and contains a putative sortase recognition motif (SPQTG) at the C-terminal end (Supplementary Fig. 4B). To investigate whether this protein is a substrate of SrtB4, we constructed plasmids expressing FLAG- or fluorescent protein-tagged RGna_12050 and introduced them into both wt and *srtB4* mutant strains. However, we did not observe cell wall localization of the fusion proteins or any apparent differences in the protein localization between the strains.

These results suggest that RGna_12050 protein is not a target of SrtB4 or that our detection system was not suitable for assessing its localization. At present, the mechanism by which *srtB4* contributes

to CPS biogenesis remains unclear. We agree that this is an important issue that should be explored in future investigations.

Fig. 6: What are the colonization levels of the two strains in the mono-colonization experiments? Could the observed differences in colitis severity be explained by differences in colonization levels?

Response: We appreciate the reviewer's insightful suggestion. We measured CFUs of both wt and *srtB4::erm* in the feces of mono-associated mice at day 0. Additionally, the number of *M. gnavus* cells was quantified by qPCR at days 0 and 8 post-DSS treatment. These values were not substantially different, as shown in revised Fig. 6C. These results suggest that the observed differences in colitis severity are likely due to the genetic characteristics of the mutant strain rather than differences in colonization levels. In the revised manuscript, we have added this description and the relevant data.

L243. "To further confirm whether CPS production is involved in intestinal inflammation induced by *M. gnavus*, we prepared *M. gnavus* wild-type or *srtB4* mutant-mono-colonized mice and then compared inflammation during administration with dextran-sodium sulfate (DSS) (Fig. 6B). Despite no substantial difference in colonization levels between wt and *srtB4* mutant in gnotobiotic mice at day 0 and a modestly increased number of *srtB4* mutant strains at 8 days post DSS treatment (Fig. 6C), we observed more severe body weight loss and increased disease activity index (DAI) in *srtB4* mutant-colonized mice compared to wild-type (Fig. 6D and E). These data suggest that genetic differences in *M. gnavus* colonizing the intestine influence inflammation, and that loss of CPSs may exacerbate its severity."

Fig. 7: The comparison of three strains is not strong evidence to link srtB4 to CPS production. Including additional strains or demonstrating gene deletion in the RI1 strain would strengthen the conclusion.

Response: Thank you for this valuable suggestion. In response, we constructed a *srtB4* homologue disruptant in the RI1 strain. This mutant strain exhibited a marked reduction in CPS presentation, as observed under a microscope with Indian-ink staining. This additional result demonstrates that the *srtB4* gene plays an important role in CPS presentation, not only in the type strain but also in other *M. gnavus* strains. We have incorporated this new data into the revised manuscript (Fig. 7B and C, L269).

Minor:

Line 52: The description linking R. gnavus to IBD severity is vague. Please clarify whether it is the abundance or presence of R. gnavus that is associated with symptoms, and whether this is a positive correlation.

Response: We replaced the sentence with "*M. gnavus* is frequently detected in the intestines of patients with these diseases, and its increased abundance is often positively correlated with the severity of symptoms. (L53)" We believe the revised sentence clarifies the correlation between *M. gnavus* abundance and symptom severity.

Line 93: please explain the term “SD” sequence.

Response: We added the abbreviation “the Shine-Dalgarno (SD) sequence (L98)”.

Fig. 1F, 2A, and others: Please indicate the expected location of the targeted proteins in the WB.

Response: We have revised the Western blotting images, including Fig. 1F, 2A, and others, to add arrows indicating the expected locations of the target proteins. We hope these revisions improve the clarity of the data.

The methods section should describe how complementation experiments corresponding to Fig. 3F and 3G were performed. These complementation strains are not listed in Table S1.

Response: We have now added a detailed description of the construction of the *srtB4* complementation plasmid to the Methods section (L387). Additionally, this plasmid has been listed in Supplementary Table S2, which provides an overview of all plasmids used in the study.

Please consider depositing the plasmids to Addgene to facilitate distribution. Including the replicon sequence in the SI table would also be helpful.

Response: We have included the replicon sequence in Supplementary Table S4, as suggested. We are currently depositing the plasmids used in this study with Addgene (Addgene Deposit number: 86940).

Line 288: Please provide more detail regarding strain dependence. In Supplementary Figure 6, many strains appear to lack the upstream genes involved in sugar biosynthesis and glycosylation. In those strains, which genes are present instead? Are they homologs of the genes in ATCC 29149, or are they entirely different genes?

Response: We appreciate the reviewer's important comment. In the revised Supplementary Fig. 8A, we reanalyzed the genomes of the *srtB4*-positive isolates using tBLASTx with more lenient criteria to identify distant homologs of the CPS gene cluster from the type strain (ATCC 29149). This analysis revealed that all *srtB4*-positive isolates possess at least one homologous gene related to sugar biosynthesis and glycosyltransferase. In addition, we added a comparison of the genetic contexts between ATCC 29149 and R11 strains, indicating that the overall gene composition and organization vary among strains (Supplementary Fig. 8B). These data support the idea that *R. gnavus* strains harbor diverse CPS gene clusters, potentially resulting in capsular polysaccharides with different sugar compositions and structures.

Reviewer #2 (Remarks to the Author):

This study begins with the construction of a suite of genetic tools that greatly enhance the genetic systems currently available to work with M. gnavus. These include an inducible gene expression system where two inducers lead to reduced leaky expression. The team also designs constructs for gene deletions and constructs for fluorescent protein analysis in M. gnavus. The bulk of the study addresses the role of various sortases in the linking of CPS and proteins to the cell using isogenic mutants. Most of these studies are well performed and provide new important data. The study includes the use of gnotobiotic mice to show fitness defects of particular mutants during competitive colonization with WT. The studies included in Figure 6 and 7 are the weakest of the study and relate to inflammatory responses or protection due to the CPS, and genetic analyses of the CPS locus in healthy and Crohn's disease patients. In total, this study significantly contributes to the field, but some areas need additional experimentation or analyses to better justify the conclusions.

Response: Thank you for your insightful comments. We have carefully revised the manuscript as the reviewer suggested, especially in Fig. 6 and 7, and have addressed each comment point-by-point. We hope the revisions improve the manuscript and meet your expectations.

The genetic tools are a significant advance and must be made available through AddGene or another not for profit plasmid distribution company.

Response: We are currently depositing the plasmids used in this study with Addgene (Addgene Deposit number: 86940).

The newly reclassified genus must be used in the title and throughout and initially the authors should write "formerly Ruminococcus gnavus".

Response: We have thoroughly revised the manuscript and have used "*Mediterraneibacter gnavus*" in the revised manuscript.

Lines 27 and 28 of the abstract should be rewritten to state that "mutations of six of the eight sortase encoding genes allowed for the identification of which are involved in the surface localization of the CPS and proteins predicted to have superantigen properties." As written, it sounds as if you show that sortases are involved in surface attachment of molecules in Gram positive bacteria, which is well established.

Response: Thank you for your careful and constructive comment. We have revised the abstract as suggested to clarify that our study identifies which sortase genes are involved in the surface localization of CPS and predicted superantigen proteins, rather than reiterating the general role of sortases in Gram-positive bacteria.

To say that the organism is a "commensal" and then say that it is associated with many diseases is not correct terminology. The term symbiont should be used or you can should say it if frequently a member of the human gut microbiota.

Response: Thank you for your valuable comment. To address this point, we have replaced the term “commensal” with “symbiont” or “symbiotic bacterium” throughout the manuscript to more accurately reflect the role of *M. gnavus* in the context of its association with various diseases.

The caveat that these aero-intolerant bacteria must be exposed to oxygen for two hours for fluorophore maturation should be discussed. It is assumed that cultures were shaken during this exposure, which if so, should be stated in the methods.

Response: As you mentioned, the maturation of fluorescent proteins requires oxygen, and we briefly described the necessity of oxygen exposure before fluorescence imaging in the main text (L105). We agitated the cell suspension using a ThermoMixer at 1400 rpm to ensure efficient oxygen exposure, and the procedure has been described in the Materials and Methods section (L427). We did not observe substantial cell lysis after 2 h exposure, supporting the applicability of this approach for labeling *M. gnavus*.

*Line 53 “This implies that *R. gnavus* could induce inflammation by blooming in the intestines.” This does not make that implication. They may just be able to better survive in inflammatory states.*

Response: We appreciate your constructive comments. As you pointed out, we do not have direct evidence that *M. gnavus* induces inflammation. It is also possible that *M. gnavus* is simply better adapted to survive and proliferate under the inflammatory conditions in the gut. We have revised the sentence accordingly to

(L55) “This implies that *M. gnavus* may induce inflammation by blooming in the intestines, or alternatively, may preferentially thrive in the inflammatory environment.”

Line 60 – change “disease activities” to “inflammation”

Response: We have revised the phrase as you suggested.

Line 71 “specific sortase enzymes”

Response: We have revised the phrase as you suggested.

The last sentence of your introduction is another obvious statement and could be better tailored to your findings

Response: We appreciate the reviewer’s constructive comments. We have revised the sentence to clarify our findings, accordingly to

(L79) “In this study, we developed a genetic toolkit that enables the functional dissection of specific sortase genes and capsular polysaccharide biosynthesis pathways in *M. gnavus*, a human gut symbiont. This work provided novel insights into how strain-specific genetic variation shapes surface structure and may influence host interactions and disease relevance.

Line 335 – remove the word “sequence”. Give a brief description of pMTL83151 “a shuttle plasmid used in Clostridiodes species”.

Response: We have removed the word “sequence” from the original text. In addition, we revised the sentence, including the description of pMTL83151, accordingly to “We PCR amplified the putative *M. gnavus* *ori* region, including the *repA* gene, from the NBRC114413 plasmid (Supplementary Table 4). We also amplified a fragment containing an antibiotic resistance gene (*catP*), *traJ*, *ColE1 ori*, and MCS from pMTL83151, a shuttle vector used in *Clostridium* species. (L381)”

*Line 108 – what is meant by two copies of the gene, *ibpA*, and *ibpB*? Do you mean one copy of each? If so, just write encoded by genes *ibpA*, and *ibpB*.*

Response: Thank you for pointing this out. We agree that the original phrasing was misleading. We have revised the sentence, as follows: “encoded by the genes *ibpA* and *ibpB*, (L117)” as we intended to indicate one copy of each gene.

Line 110 – Did you do a whole genome sequence of this strain? This is now quite inexpensive and is necessary to show that there are not off-target effects from the CRISPR/Cas system.

Response: Thank you for your suggestion. We performed whole-genome sequencing of these strains as you suggested and added a description stating that there are no off-target effects from the CRISPR/Cas system (L120).

Line 113- better to write “other members of the Lachnospiraceae family”.

Response: We revised the phrase as the reviewer suggested.

*Line 121 – give your rationale for naming two of the sortase genes *srtA* and 6 of them *srtB*.*

Response: In the revised manuscript, we have added a phylogenetic analysis of sortase proteins derived from *R. gnavus* and various other Gram-positive bacteria (supplementary Fig. S4A). This analysis revealed that most *M. gnavus* sortase B homologs (RGna_01825, RGna_09760, RGna_10250, RGna_12035, RGna_13420) are phylogenetically close to known sortase B proteins in other species. Additionally, we found that RGna_09235 and RGna_13410, which were initially annotated as *SrtA*, clustered with the sortase C proteins in the phylogenetic tree. Based on this observation, we have renamed these genes as *srtC1* and *srtC2*, respectively. We have revised the original sentence in the revised manuscript, as follows: “*M. gnavus* ATCC 29149, a type strain, has eight putative sortase genes, which we named *srtB1-6* and *srtC1-2* based on a phylogenetic analysis of their amino acid sequences (Supplementary Fig. S4). (L131)”

Line 130 – maybe better to say “in sortase-processed cell wall anchored proteins”

Response: We have revised the phrase as you suggested.

Figure S3C –As you have a genome sequence, did it reveal two copies of this gene cluster? This should be performed with long read technology to overcome problems assembling repetitive regions. Is this region likely a mobile genetic element? If so, these srt genes may not be present in present in all M. gnavus strains. MGEs often insert into tRNA genes, which you could search for at the junctions of the duplicated region. What other genes are present in this duplicated region?

Response: We have observed a ~2-fold increase in read depth across the locus spanning RGna_13305 to RGna_13530, strongly suggesting that this region is duplicated in the genome. However, as you pointed out, because we used short-read sequencing for genome resequencing, we cannot determine the precise location and orientation of this duplicated region.

In addition, we identified a putative transposase gene in this duplicated region, immediately upstream of tRNA genes, suggesting that this region may represent an MGE. We have revised Supplementary Fig. S5 to show the location of a putative transposase gene and the tRNA genes. These findings imply that the presence of *srtC2* and *srtB6* could be strain-dependent.

Line 148 – better to say “formed a less compact pellet”

Response: We have revised the phrase as you suggested.

Fig 3B and 3F must show a MW range so that we can determine the size of the CPS. They are typically quite heterogeneously sized high MW molecules.

Response: We have revised Fig. 3B and 3F to include molecular weight markers, as you suggested. We used a pre-stained protein ladder and enlarged the cropped regions to show a broader molecular weight range of the SDS-PAGE images. We detected CPS bands above the largest molecular weight marker, suggesting they are high-molecular-weight polysaccharides, consistent with the typical characteristics of CPS.

Fig 3C – are there any other phenotypes of the srtB4 mutant? In the image, it appears to have shorter chains. Do these bacteria clump at all as is a characteristic of some bacteria when their capsules are removed.

Response: Thank you for your comment. Based on microscopic observation, the chain length of the *srtB4* mutant appears comparable to that of the wt strain in our experimental conditions. Representative phase contrast images of both strains at the exponential phase are shown below.

Supporting information 2. Phase contrast microscopy of wt and $\Delta srtB4$. Cells were grown at the exponential phase (O.D.600 \approx 1.0). Bars = 10 μ m.

Line 152-154 – this image is not enough proof to say that these are fimbriae. These bacteria were processed for EM which introduces all types of artifacts. This statement about fimbriae should be removed.

Response: To avoid overinterpretation, we have replaced “fimbriate-like” with “fibrous” to state only the morphology observed in the TEM images, as you suggested.

The figure legend of Fig 3 (and line 161) should be modified, *srtB4* is not responsible for capsular polysaccharide production, it is likely important in linking the CPS to the bacterial surface. Did you analyze the supernatants for the presence of CPS? An antibody to this molecule would substantially help with analyses – which should be easily obtained by adsorbing antibodies to the WT bacterium with the CPS mutant.

Response: Thank you for this important and thoughtful comment. We agree that our original statement may have overinterpreted the role of SrtB4. Based on our data, we observed a substantial reduction of CPS structures on the cell surface in the *srtB4* mutant, as assessed by microscopy and SDS-PAGE. We analyzed the CPS in the supernatants and found that the amount of the CPS is not greatly changed between the wt and *srtB4* mutant strains (see below), suggesting that reduction of the CPS in the mutant is not simply explained by the release into the medium.

Supporting information 3. Detection of CPS in the supernatants. Capsular polysaccharide fractions (C) and supernatants (S) collected from cells grown overnight were separated by SDS-PAGE. Polysaccharides were detected by acid polysaccharide staining dye, alcian blue. Each lane contains 0.5 O.D. 600 units.

However, at this point, we cannot determine whether *srtB4* affects CPS production, surface anchoring, or both. To avoid overinterpretation, we have revised the description to state *srtB4* is responsible for the surface presentation of CPS.

Regarding your suggestion to generate anti-CPS antibodies, we appreciate this valuable idea and agree that such a tool would facilitate more detailed analysis. However, due to time constraints and resource limitations, we were unable to generate such antibodies in the current study. We hope to pursue this approach in future work.

Line 167 –state “competitive advantage of the WT strains against the sortase mutants”, rather than

the other way around.

Response: We have revised the phrase as you suggested.

Lines 200-201 – This statement is not supported as written. Bacteria frequently make more than one CPS/EPS. It is quite likely that the locus identified in the other study also produces CPS that is not the same as the CPS of your study. Many CPS do not extend so far from the cell and are not nearly as visible with India ink staining as the CPS you identified. This statement should be qualified to say “these gene disruptions did not affect the phenotypes tested in our study”. In fact, the cell pellets of these deletion mutants seem to be even more loosely packed than the WT strain, suggesting some role in production or presentation of surface PSs.

Response: Thank you for your thoughtful comment. We agree that our original statement was overinterpreted, and we did not intend to exclude the possibility that the genes identified in the previous study may be involved in the biosynthesis of polysaccharides other than those we tested in this study. Accordingly, we have revised the sentence as you suggested.

In addition, we appreciate your insightful observation that these mutants showed the looser cell pellets. Although we did not quantify this phenotype under our experimental conditions, we agree that this implies the involvement of these genes in surface polysaccharide production or presentation. We consider exploring it as an intriguing direction of future studies.

Line 204 – there is no evidence that SrtB4 affects CPS production, just likely its surface attachment, or the attachment of another protein that is necessary for its surface localization.

Response: As mentioned earlier, we agree that our original statement may have overinterpreted the role of SrtB4. Based on our data, we cannot determine whether *srtB4* affects CPS production, surface anchoring, or both. To avoid overinterpretation, we have revised the description to state *srtB4* is responsible for CPS presentation to cell surfaces.

There are some concerns with the CPS mutants analyzed for competitive colonization in the mouse gut. As gene 11905 encodes a nucleotide-sugar synthesis gene, it may affect the production of nucleotide linked sugars necessary for other cell glycans. However, there are also concerns with any mutants that allow the incomplete building or flipping and polymerization of the repeat unit, as these mutants sequester and prevent recycling of undecaprenol-P, which is necessary for PG synthesis and can contribute to growth defects. The best strategy is to delete all glycosyltransferase genes so that the repeat unit is not built at all. These caveats should at least be discussed for the competitive colonization assays, especially since the strains have an in vitro growth defect. However, the 12040 mutant, which based on gene proximity may only be involved in attaching the CPS (although that is unknown), also confirms the competitive defect.

Response: Thank you for your thoughtful comment. As you suggested, we cannot exclude the possibility that the inability to produce nucleotide sugars influences a growth defect other than

polysaccharide surface presentation and causes decreased competitive fitness in the intestine. We have revised the paragraph that describes the competitive colonization assay of RGna_11905 and RGna_12040 mutants, as follows:

(L226) "Because RGna_11905 is a putative nucleotide-sugar synthesis gene, its disruption may affect the production of nucleotide-linked sugars necessary for other cell glycans beyond CPS. This possibly contributes to the observed decrease in competitive fitness. However, the competitive defect was also observed in the RGna_12040 mutant, which is likely involved in CPS attachment rather than synthesis. These results are likely to rule out a simple difference in growth rate as the cause of the CPS mutant strains being outcompeted by wild-type in the intestinal environment. Taken together, CPS production and surface localization are important for competitive fitness during intestinal colonization in *M. gnavus*."

We believe these revised sentences clarify and address the caveats raised by the reviewer.

Lines 215-216 – "The CPS produced by the gene products of the identified locus....". There may be more than one CPS/EPS produced by this organism.

Response: We agree with the reviewer's suggestion. We have revised the phrase as you suggested.

Line 220 – this statement is too broad, and the opposite has also been shown. Maybe say "to have various roles in immune recognition or modulation".

Response: Thank you for your suggestion. We have revised the sentence as you suggested.

*Fig 6 – there is some concern for the very different amounts of TNF in the strains comparing panels A and B. Why is there such a difference in the WT and *srtB::erm* strains in these panels?*

Response: We acknowledge the difference in TNF concentrations between Fig. 6A and B in the original figure. In our experimental setting, these variations were occasionally observed even within the wt strain group across three independent experiments. Nevertheless, we consistently observed the same tendency across the experiments: *srtB4* or CPS mutants induced higher TNF production than the wt strains. To avoid potential confusion, we have replaced the original Fig. 6A and B with a new Fig. 6A that shows representative data from a single experiment, including all relevant strains.

*Figure 6C – it is not evident when these mice were gavaged, please put an arrow on the time line. Also, it is surprising that mice monocolonized with *M. gnavus* survived for 8 days on 2.5% DSS.*

Response: We have revised the original Fig. 6C to include arrows indicating the time point of oral gavage, as you suggested. Regarding the DSS concentration, BALB/c mice are reported to be more tolerant to DSS treatment than C57BL/6 mice [1], which likely explains the survival of *M. gnavus*-mono-associated mice for 8 days in our experiment.

[1] Chassaing, Benoit, et al. "Dextran sulfate sodium (DSS)-induced colitis in mice." Current protocols

in immunology 104.1 (2014): 15-25.

Fig 6D – for this experiment, you must quantify the cfu/gram of bacteria in these mice over the course of the experiment (for a few time points). It is very possible that the mutant strain was not able to colonize the DSS treated mouse to the same extent as the WT, which would be relevant data. Based on the data shown, I disagree with the statement “These results suggest that CPS production inhibits the recognition of R. gnavus cells from host immune cells and that CPS233 deficient R. gnavus has the potential to induce more severe inflammation during intestinal colonization”. The inflammation is induced by the DSS, not the M. gnavus. Without cfu/g, there cannot be a statement about effects of the CPS in this assay.

Response: We have revised Fig. 6 to include CFUs/g feces and the number of cells calculated by qPCR for each strain, as you suggested. These data indicate that colonization levels were not substantially different between the wt and mutant strains. Thus, the observed difference in DSS-induced inflammation is not due to differences in colonization efficiency but rather to genetic differences between these strains. We also replaced the last sentence of this paragraph as follows:

(L243) “To further confirm whether CPS production is involved in intestinal inflammation induced by *M. gnavus*, we prepared *M. gnavus* wild-type or *srtB4* mutant-mono-colonized mice and then compared inflammation during administration with dextran-sodium sulfate (DSS) (Fig. 6B). Despite no substantial difference in colonization levels between wt and *srtB4* mutant in gnotobiotic mice at day 0 and a modestly increased number of *srtB4* mutant strains at 8 days post DSS treatment (Fig. 6C), we observed more severe body weight loss and increased disease activity index (DAI) in *srtB4* mutant-colonized mice compared to wild-type (Fig. 6D and E). These data suggest that genetic differences in *M. gnavus* colonizing the intestine influence inflammation, and that loss of CPSs may exacerbate its severity.”

Studies related to figure 7 are problematic

1) are you sure that all of these strains were isolated from separate people and are not replicates (possible longitudinal samples) from the same individual? In addition, the geographic location of the isolates (from where they live) is necessary as there can be clustering of isolates in different countries which would confound the data. There is some concern for the low number of samples for this analysis, which may be lower if duplicate samples from the same person are revealed.

Response: We carefully reviewed the literature [2] describing the isolated strains used in this analysis. While there was no clear statement whether all strains were isolated from different individuals, available clinical metadata, such as age, sex, and nationality, suggest that most strains were likely derived from different individuals.

To address your concern, we have revised Fig. 7A to include the geographic locations of each isolate. These data suggest that there is no strong geographic bias in *srtB4* possession across countries. Despite the small sample size, we believe the trends of association between disease and the *srtB4*

gene remain informative. In addition, we have added a description of the limitations of this analysis, due to sample size, in the Discussion section.

(L303) "Additionally, the *srtB4* gene, which is required for CPS presentation, appears to be less prevalent in isolates from Crohn's disease patients than in those from healthy individuals in the available complete genome dataset, although the sample size in this study is limited. This highlights a key limitation of comparative genomics alone and underscores the importance of functional validation using a bacterium-specific genetic modification system. Nevertheless, due to the limited number of isolates with complete genome sequences, the association between *srtB4* and diseases should be interpreted with caution and warrants further investigation."

[2] Nooij, Sam, et al. "Metagenomic global survey and in-depth genomic analyses of *Ruminococcus gnavus* reveal differences across host lifestyle and health status." *Nature Communications* 16.1 (2025): 1182.

2) CPS loci are often heterogeneous across strains of a species. Do the strains that do not have the same CPS locus as described in this study have a distinct CPS locus in this region? In Fig S6, please show gene maps of this region in other strains.

Response: In the revised Supplementary Fig. 8A, we reanalyzed the genomes of the *srtB4*-positive isolates using tBLASTx with more lenient criteria to identify distant homologs of the CPS gene cluster from the type strain (ATCC 29149). This analysis revealed that all *srtB4*-positive isolates possess at least one homologous gene related to sugar biosynthesis and glycosyltransferase activity. In addition, we added a comparison of the genetic contexts of ATCC 29149, R11, and NBRC 114413 strains, indicating that the overall gene composition and organization vary among strains (Supplementary Fig. 8B). These data support the idea that *R. gnavus* strains harbor diverse CPS gene clusters, potentially resulting in capsular polysaccharides with different sugar compositions and structures.

3) it is quite possible that another Srt can serve the function of SrtB4 if there are diverse CPS loci in this region.

Response: In the revised Supplementary Fig. 8B, we include a genetic map of the corresponding CPS locus in NBRC 114413, a *srtB4*-negative strain. This region contains several putative CPS-related genes (e.g., glycosyltransferases, sugar-nucleoside biosynthesis, and flippases) but lacks sortase genes. Similarly, none of the *srtB4*-negative strains used in this study possess sortase genes at the corresponding locus. These suggest that, at least within this CPS locus, it is unlikely that another sortase gene is substituted for *srtB4* function.

Statements in the discussion related to findings of Crohn's disease patients are overstated based on the caveats mentioned above.

Response: As noted earlier, we acknowledge the potential overstatement regarding the association

between *srtB4* and diseases. Therefore, we have revised the Discussion section to tone down the statements as previously described.

REVIEWERS' COMMENTS

Reviewer #1 (Remarks to the Author):

The revised manuscript shows improved quality, and I now have only a minor comment.

At Lines 120 and 285–295, the associated data (genomic analysis, plasmid loss, and transformation of other species) should be included in the supplementary materials.

Response: We thank the reviewer for the positive evaluation. We have included data from genomic analysis in Supplementary Fig. 3f, which shows no off-target effects. Additionally, we included images showing the loss of the CRISPR plasmid and the fluorescent strains of *E. bolteae* and *E. clostridioformis* in the new Supplementary Fig. 8.